# Implicit Gradient Regularization

**David G.T. Barrett**[*]
DeepMind
London
barrettdavid@google.com

**Benoit Dherin**[*]
Google
Dublin
dherin@google.com

## Abstract

Gradient descent can be surprisingly good at optimizing deep neural networks without overfitting and without explicit regularization. We find that the discrete steps of gradient descent implicitly regularize models by penalizing gradient descent trajectories that have large loss gradients. We call this *Implicit Gradient Regularization* (IGR) and we use backward error analysis to calculate the size of this regularization. We confirm empirically that implicit gradient regularization biases gradient descent toward flat minima, where test errors are small and solutions are robust to noisy parameter perturbations. Furthermore, we demonstrate that the implicit gradient regularization term can be used as an explicit regularizer, allowing us to control this gradient regularization directly. More broadly, our work indicates that backward error analysis is a useful theoretical approach to the perennial question of how learning rate, model size, and parameter regularization interact to determine the properties of overparameterized models optimized with gradient descent.

## 1 Introduction

The loss surface of a deep neural network is a mountainous terrain - highly non-convex with a multitude of peaks, plateaus and valleys (Li et al., 2018; Liu et al., 2020). Gradient descent provides a path through this landscape, taking discrete steps in the direction of steepest descent toward a sub-manifold of minima. However, this simple strategy can be just as hazardous as it sounds. For small learning rates, our model is likely to get stuck at the local minima closest to the starting point, which is unlikely to be the most desirable destination. For large learning rates, we run the risk of ricocheting between peaks and diverging. However, for moderate learning rates, gradient descent seems to move away from the closest local minima and move toward flatter regions where test data errors are often smaller (Keskar et al., 2017; Lewkowycz et al., 2020; Li et al., 2019). This phenomenon becomes stronger for larger networks, which also tend to have a smaller test error (Arora et al., 2019a; Belkin et al., 2019; Geiger et al., 2020; Liang & Rakhlin, 2018; Soudry et al., 2018). In addition, models with low test errors are more robust to parameter perturbations (Morcos et al., 2018). Overall, these observations contribute to an emerging view that there is some form of implicit regularization in gradient descent and several sources of implicit regularization have been identified.

We have found a surprising form of implicit regularization hidden within the discrete numerical flow of gradient descent. Gradient descent iterates in discrete steps along the gradient of the loss, so after each step it actually steps off the exact continuous path that minimizes the loss at each point. Instead of following a trajectory down the steepest local gradient, gradient descent follows a shallower path. We show that this trajectory is closer to an exact path along a modified loss surface, which can be calculated using backward error analysis from numerical integration theory (Hairer et al., 2006). Our core idea is that the discrepancy between the original loss surface and this modified loss surface is a form of implicit regularization (Theorem 3.1, Section 3).

We begin by calculating the discrepancy between the modified loss and the original loss using backward error analysis and find that it is proportional to the second moment of the loss gradients, which we call *Implicit Gradient Regularization* (IGR). Using differential geometry, we show that IGR is also proportional to the square of the loss surface slope, indicating that it encourages optimization paths with shallower slopes and optima discovery in flatter regions of the loss surface. Next, we

---

[*]equal contribution

explore the properties of this regularization in deep neural networks such as MLP's trained to classify MNIST digits and ResNets trained to classify CIFAR-10 images and in a tractable two-parameter model. In these cases, we verify that IGR effectively encourages models toward minima in the vicinity of small gradient values, in flatter regions with shallower slopes, and that these minima have low test error, consistent with previous observations. We find that IGR can account for the observation that learning rate size is correlated with test accuracy and model robustness. Finally, we demonstrate that IGR can be used as an explicit regularizer, allowing us to directly strengthen this regularization beyond the maximum possible implicit gradient regularization strength.

## 2    THE MODIFIED LOSS LANDSCAPE INDUCED BY GRADIENT DESCENT

The general goal of gradient descent is to find a weight vector $\hat{\theta}$ in parameter space $\mathbb{R}^m$ that minimizes a loss $E(\theta)$. Gradient descent proceeds by iteratively updating the model weights with learning rate $h$ in the direction of the steepest loss gradient:

$$\theta_{n+1} = \theta_n - h\nabla_\theta E(\theta_n) \tag{1}$$

Now, even though gradient descent takes steps in the direction of the steepest loss gradient, it does not stay on the exact continuous path of the steepest loss gradient, because each iteration steps off the exact continuous path. Instead, we show that gradient descent follows a path that is closer to the exact continuous path given by $\dot{\theta} = -\nabla_\theta \widetilde{E}(\theta)$, along a modified loss $\widetilde{E}(\theta)$, which can be calculated analytically using backward error analysis (see Theorem 3.1 and Section 3), yielding:

$$\widetilde{E}(\theta) = E(\theta) + \lambda R_{IG}(\theta), \tag{2}$$

where

$$\lambda \equiv \frac{hm}{4} \tag{3}$$

and

$$R_{IG}(\theta) \equiv \frac{1}{m} \sum_{i=1}^{m} \left(\nabla_{\theta_i} E(\theta)\right)^2 \tag{4}$$

Immediately, we see that this modified loss is composed of the original training loss $E(\theta)$ and an additional term, which we interpret as a regularizer $R_{IG}(\theta)$ with regularization rate $\lambda$. We call $R_{IG}(\theta)$ the *implicit gradient regularizer* because it penalizes regions of the loss landscape that have large gradient values, and because it is implicit in gradient descent, rather than being explicitly added to our loss.

**Definition.** *Implicit gradient regularization is the implicit regularisation behaviour originating from the use of discrete update steps in gradient descent, as characterized by Equation 2.*

We can now make several predictions about IGR which we will explore in experiments:

**Prediction 2.1.** *IGR encourages smaller values of $R_{IG}(\theta)$ relative to the loss $E(\theta)$.*

Given Equation 2 and Theorem 3.1, we expect gradient descent to follow trajectories that have relatively small values of $R_{IG}(\theta)$. It is already well known that gradient descent converges by reducing the loss gradient so it is important to note that this prediction describes the *relative* size of $R_{IG}(\theta)$ along the trajectory of gradient descent. To expose this phenomena in experiments, great care must be taken when comparing different gradient descent trajectories. For instance, in our deep learning experiments, we compare models at the iteration time of maximum test accuracy (and we consider other controls in the appendix), which is an important time point for practical applications and is not trivially determined by the speed of learning (Figures 1, 2). Also, related to this, since the regularization rate $\lambda$ is proportional to the learning rate $h$ and network size $m$ (Equation 3), we expect that larger models and larger learning rates will encourage smaller values of $R_{IG}(\theta)$ (Figure 2).

**Prediction 2.2.** *IGR encourages the discovery of flatter optima.*

In section 3 we will show that $R_{IG}(\theta)$ is proportional to the square of the loss surface slope. Given this and Prediction 2.1, we expect that IGR will guide gradient descent along paths with shallower loss surface slopes, thereby encouraging the discovery of flatter, broader optima. Of course, it is possible to construct loss surfaces at odds with this (such as a Mexican-hat loss surface, where all minima are equally flat). However, we will provide experimental support for this using loss surfaces that are of widespread interest in deep learning, such as MLPs trained on MNIST (Figure 1, 2, 3).

**Prediction 2.3.** *IGR encourages higher test accuracy.*

Given Prediction 2.2, we predict that IGR encourages higher test accuracy since flatter minima are known empirically to coincide with higher test accuracy (Figure 2).

**Prediction 2.4.** *IGR encourages the discovery of optima that are more robust to parameter perturbations.*

There are several important observations to make about the properties of IGR: 1) It does not originate in any specific model architecture or initialization, although our analysis does provide a formula to explain the influence of these model properties *through* IGR; 2) Other sources of implicit regularization also have an impact on learning, alongside IGR, and the relative importance of these contributions will likely depend on model architecture and initialization; 3) In defining $\lambda$ and $R_{IG}$ we chose to set $\lambda$ proportional to the number of parameters $m$. To support this choice, we demonstrate in experiments that the test accuracy is controlled by the IGR rate $\lambda$. 4) The modified loss and the original loss share the same global minima, so IGR vanishes when the gradient vanishes. Despite this, the presence of IGR has an impact on learning since it changes the *trajectory* of gradient descent, and in over-parameterized models this can cause the final parameters to reach different solutions. 5) Our theoretical results are derived for full-batch gradient descent, which allows us to isolate the source of implicit regularisation from the stochasticity of stochastic gradient descent (SGD). Extending our theoretical results to SGD is considerably more complicated, and as such, is beyond the scope of this paper. However, in some of our experiments, we will demonstrate that IGR persists in SGD, which is especially important for deep learning. Next, we will provide a proof for Theorem 3.1, and we will provide experimental support for our predictions.

## 3 BACKWARD ERROR ANALYSIS OF GRADIENT DESCENT

In this section, we show that gradient descent follows the gradient flow of the modified loss $\widetilde{E}$ (Equation 2) more closely than that of the original loss $E$. The argument is a standard argument from the backward error analysis of Runge-Kutta methods (Hairer et al., 2006). We begin by observing that gradient descent (Equation 1) can be interpreted as a Runge-Kutta method numerically integrating the following ODE:

$$\dot{\theta} = -\nabla_\theta E(\theta) \tag{5}$$

In the language of numerical analysis, gradient descent is the explicit Euler method numerically integrating the vector field $f(\theta) = -\nabla E(\theta)$. The explicit Euler method is of order 1, which means that after one gradient descent step $\theta_n = \theta_{n-1} - h\nabla E(\theta_{n-1})$, the deviation from the gradient flow $\|\theta_n - \theta(h)\|$ is of order $\mathcal{O}(h^2)$, where $\theta(h)$ is the solution of Equation 5 starting at $\theta_{n-1}$ and evaluated at time $h$. Backward error analysis was developed to deal with this discrepancy between the discrete steps of a Runge-Kutta method and the continuous exact solutions (or flow) of a differential equation. The main idea is to modify the ODE vector field $\dot{\theta} = f(\theta)$ with corrections in powers of the step size

$$\tilde{f}(\theta) = f(\theta) + hf_1(\theta) + h^2 f_2(\theta) + \cdots \tag{6}$$

so that the numerical steps $\theta_n$ approximating the original Equation 5 now lie exactly on the solutions of the *modified equation* $\dot{\theta} = \tilde{f}(\theta)$. In other words, backward error analysis finds the corrections $f_i$ in Equation 6 such that $\theta_n = \tilde{\theta}(nh)$ for all $n$, where $\tilde{\theta}(t)$ is the solution of the modified equation starting at $\theta_0$. In theory, we can now precisely study the flow of the modified equation to infer properties of the numerical method because its steps follow the modified differential equation solutions perfectly in a formal sense. The following result is a direct application of backward error analysis to gradient descent:

**Theorem 3.1.** *Let $E$ be a sufficiently differentiable function on a parameter space $\theta \in \mathbb{R}^m$. The modified equation for gradient flow (Equation 5) is of the form*

$$\dot{\theta} = -\nabla\widetilde{E}(\theta) + \mathcal{O}(h^2) \tag{7}$$

*where $\widetilde{E} = E + \lambda R_{IG}$ is the modified loss introduced in Equation 2. Consider gradient flow with the modified loss $\dot{\theta} = -\nabla\widetilde{E}(\theta)$ and its solution $\tilde{\theta}(t)$ starting at $\theta_{n-1}$. Now the local error $\|\theta_n - \tilde{\theta}(h)\|$ between $\tilde{\theta}(h)$ and one step of gradient descent $\theta_n = \theta_{n-1} - h\nabla E(\theta_{n-1})$ is of order $\mathcal{O}(h^3)$, while it is of order $\mathcal{O}(h^2)$ for gradient flow with the original loss.*

*Proof.* We begin by computing $f_1$ for which the first two orders in $h$ of the Taylor's Series of the modified equation solution $\theta(t)$ at $t = h$ coincide with one gradient descent step. Since $\theta'(t) = \tilde{f}(\theta)$, we see that $\theta''(t) = \tilde{f}'(\theta)\tilde{f}(\theta)$ and we find that

$$\theta + hf(\theta) = \theta + hf(\theta) + h^2\big(f_1(\theta) + \frac{1}{2}f'(\theta)f(\theta)\big),$$

yielding $f_1(\theta) = -f'(\theta)f(\theta)/2$. Now, when $f$ is a gradient vector field with $f = -\nabla E$, we find:

$$f_1(\theta) = -\frac{1}{2}(D_\theta^2 E)\nabla_\theta E(\theta) = -\frac{1}{4}\nabla\|\nabla E(\theta)\|^2,$$

where $D_\theta^2 E$ is the Hessian of $E(\theta)$. Putting this together, we obtain the first order modified equation:

$$\dot{\theta} = f + hf_1 + \mathcal{O}(h^2) = -\nabla\Big(E(\theta) + \frac{h}{4}\|\nabla E(\theta)\|^2\Big) + \mathcal{O}(h^2),$$

which is a gradient system with modified loss

$$\widetilde{E}(\theta) = E(\theta) + \frac{h}{4}\|\nabla E(\theta)\|^2.$$

As for the local error, if $\theta(h)$ is a solution of gradient flow starting at $\theta_{n-1}$, we have in general that $\theta(h) = \theta_n + \mathcal{O}(h^2)$. The correction $f_1$ is constructed so that it cancels out the $\mathcal{O}(h^2)$ term in the expansion of its solution, yielding $\tilde{\theta}(h) = \theta_n + \mathcal{O}(h^3)$. $\qquad\square$

**Remark 3.2.** *A direct application of a standard result in backward error analysis (Hairer & Lubich (1997), Thm. 1) indicates that the learning rate range where the gradient flow of the modified loss provides a good approximation of gradient descent lies below $h_0 = CR/M$, where $\nabla E$ is analytic and bounded by M in a ball of radius R around the initialization point and where C depends on the Runge-Kutta method only, which can be estimated for gradient descent. We call this the moderate learning rate regime. For each learning rate below $h_0$, we can provably find an optimal truncation of the modified equation whose gradient flow is exponentially close to the steps of gradient descent, so the higher term corrections are likely to contribute to the dynamics. Given this, we see that the exact value of the upper bound for the moderate regime will correspond to a setting where the optimal truncation is the first order correction only. Calculating this in general is difficult and beyond the scope of this paper. Nonetheless, our experiments strongly suggest that this moderate learning rate regime overlaps substantially with the learning rate range typically used in deep learning.*

Next, we give a purely geometric interpretation of IGR, supporting Prediction 2.2. Consider the loss surface $S$ associated with a loss function $E$ defined over the parameter space $\theta \in \mathbb{R}^m$. This loss surface is defined as the graph of the loss: $S = \{(\theta, E(\theta)) : \theta \in \mathbb{R}^m\} \subset \mathbb{R}^{m+1}$. We define $\alpha(\theta)$ to be the angle between the tangent space $T_\theta S$ to $S$ at $\theta$ and the parameter plane, i.e., the linear subspace $\{(\theta, 0) : \theta \in \mathbb{R}^m\}$ in $\mathbb{R}^{m+1}$. We can compute this angle using the inner product between the normal vector $N(\theta)$ to $S$ at $\theta$ and the normal vector $\hat{z}$ to the parameter plane: $\alpha(\theta) = \arccos\langle N(\theta), \hat{z}\rangle$. Now we can define the *loss surface slope* at $\theta$ as being the tangent of this angle: $\mathrm{slope}(\theta) := \tan\alpha(\theta)$. This is a natural extension of the 1-dimensional notion of slope. With this definition, we can now reformulate the modified loss function in a purely geometric fashion:

**Proposition 3.3.** *The modified loss $\tilde{E}$ in Equation 2 can be expressed in terms of the loss surface slope as $\tilde{E}(\theta) = E(\theta) + \frac{h}{4}\mathrm{slope}^2(\theta)$.*

This proposition is an immediate consequence of Theorem 3.1 and Corollary A.6.1 in Appendix A.2. It tells us that gradient descent with higher amounts of implicit regularization (higher learning rate) will implicitly minimize the loss surface slope locally along with the original training loss. Prediction 2.2 claims that this local effect of implicit slope regularization accumulates into the global effect of directing gradient descent trajectories toward global minima in regions surrounded by shallower slopes - toward flatter (or broader) minima.

**Remark 3.4.** *It is important to note that IGR does not help gradient descent to escape from local minima. In the learning rate regime where the truncated modified equation gives a good approximation for gradient descent, the steps of gradient descent follow the gradient flow of the modified loss closely. As Proposition A.10 shows, the local minima of the original loss are still local minima of the modified loss so gradient descent within this learning rate regime remains trapped within the basin of attraction of these minima. IGR does not lead to an escape from local minima, but instead, encourages a shallower path toward flatter solutions close to the submanifold of global interpolating minima, which the modified loss shares with the original loss (Proposition A.10).*

## 4 EXPLICIT GRADIENT REGULARIZATION

For overparameterized models, we predict that the strength of IGR relative to the original loss can be controlled by increasing the learning rate $h$ (Prediction 2.1). However, gradient descent becomes unstable when the learning rate becomes too large. For applications where we wish to increase the strength of IGR beyond this point, we can take inspiration from implicit gradient regularization to motivate *Explicit Gradient Regularization* (EGR), which we define as:

$$E_\mu(\theta) = E(\theta) + \mu\|\nabla E(\theta)\|^2 \tag{8}$$

where, $\mu$ is the explicit regularization rate, which is a hyper-parameter that we are free to choose, unlike the implicit regularization rate $\lambda$ (Equation 3) which can only by controlled indirectly. Now, we can do gradient descent on $E_\mu$ with small learning rates and large $\mu$.

Although EGR is not the primary focus of our work, we will demonstrate the effectiveness of EGR for a simple two parameter model in the next section (Section 5) and for a ResNet trained on Cifar-10 (Figure 3c). Our EGR experiments act as control study in this work, to demonstrate that the $R_{IG}$ term arising implicitly in gradient descent can indeed improve test accuracy independent of confounding effects that may arise when we control IGR implicitly through the learning rate. Namely, if we had not observed a significant boost in model test accuracy by adding the $R_{IG}$ term explicitly, our prediction that implicit regularization helps to boost test accuracy would have been in doubt.

**Related work**: Explicit regularization using gradient penalties has a long history. In early work, Drucker & Le Cun (1992) used a gradient penalty (using input gradients instead of parameter gradients). Hochreiter & Schmidhuber (1997) introduced a regularization penalty to guide gradient descent toward flat minima. EGR is also reminiscent of other regularizers such as dropout, which similarly encourages robust parameterizations (Morcos et al., 2018; Srivastava et al., 2014; Tartaglione et al., 2018). More recently, loss gradient penalties have been used to stabilize GAN training (Nagarajan & Kolter, 2017; Balduzzi et al., 2018; Mescheder et al., 2017; Qin et al., 2020). The success of these explicit regularizers demonstrates the importance of this type of regularization in deep learning.

## 5 IGR AND EGR IN A 2-D LINEAR MODEL

In our first experiment we explore implicit and explicit gradient regularization in a simple two-parameter model with a loss given by $E(a, b) = (y - f(x; a, b))^2/2$, where $f(x; a, b) = abx$ is our model, $a, b \in \mathbb{R}$ are the model parameters and $x, y \in \mathbb{R}$ is the training data. We have chosen this model because we can fully visualize the gradient descent trajectories in 2-d space. For a single data point, this model is overparameterized with global minima located along a curve attractor defined by the hyperbola $ab = y/x$. For small learning rates, gradient descent follows the gradient flow of the loss from an initial point $(a_0, b_0)$ toward the line attractor. For larger learning rates, gradient descent follows a longer path toward a different destination on the line attractor (Figure 1a).

We can understand these observations using Theorem 3.1, which predicts that gradient descent is closer to the modified flow given by $\dot{a} = -\nabla_a \widetilde{E}(a, b)$ and $\dot{b} = -\nabla_b \widetilde{E}(a, b)$ where $\widetilde{E}(a, b) = E(a, b) + \lambda R_{IG}(a, b)$ is the modified loss from Equation 2, $R_{IG}(a, b) = (|a|^2 + |b|^2)x^2 E(a, b)$ is the implicit regularization term from Equation 4 and $\lambda = h/2$ is the implicit regularization rate from Equation 3, with learning rate $h$. Although the modified loss and the original loss have the same global minima, they generate different flows. Solving the modified flow equations numerically starting from the same initial point $(a_0, b_0)$ as before, we find that the gradient descent trajectory is closer to the modified flow than the exact flow, consistent with Theorem 3.1 (Figure 1a).

Next, we investigate Prediction 2.1, that the strength of the implicit gradient regularization $R_{IG}(a, b)$ relative to the original loss $E(a, b)$ can be controlled by increasing the regularization rate $\lambda$. In this case, this means that larger learning rates should produce gradient descent trajectories that lead to minima with a smaller value of $R_{IG}(a, b)/E(a, b) = x^2(|a|^2 + |b|^2)$. It is interesting to note that this is proportional to the parameter norm, and also, to the square of the loss surface slope. In our numerical experiments, we find that larger learning rates lead to minima with smaller L2 norm (Figure 1b), closer to the flatter region in the parameter plane, consistent with Prediction 2.1 and 2.2. The extent to which we can strengthen IGR in this way is restricted by the learning rate. For excessively

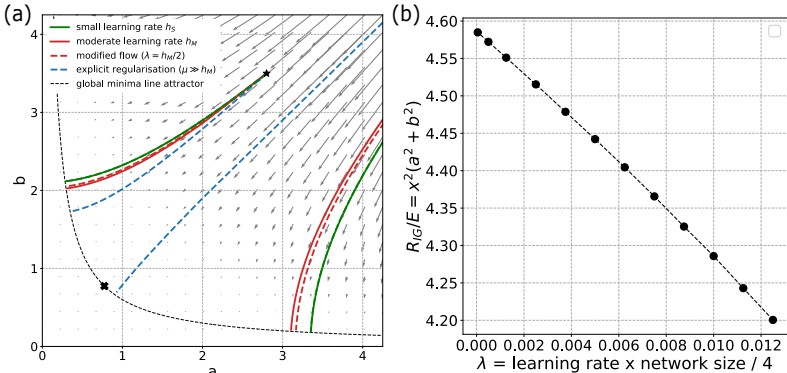

Figure 1: Implicit gradient regularization and explicit gradient regularization for a simple 2-d model. (a) In this phase space plot, the loss surface of a simple overparameterized model is represented with arrows denoting the direction of gradient flow from each point, with global minima represented by a line attractor (black dashed line). Gradient descent trajectories originating from two different starting points are illustrated (from $a_0 = 2.8, b_0 = 3.5$, and from $a_0 = 75, b_0 = 74.925$, off plot). The global minima in the flattest region of the loss surface and with lowest L2 norm is indicated with a black cross. For small learning rate, $h_S$, gradient descent trajectories follow the exact gradient flow closely (green lines). For a moderate learning rate $h_M$, gradient descent follows a longer trajectory (red lines). This is closer to the corresponding exact flow following the modified loss gradient, with $\lambda = h_M/2$, consistent with backward error analysis. We also calculate the gradient descent trajectory with explicit regularization (dashed blue lines) using large regularization rates ($\mu \gg h_M$). (b) The ratio $R_{IG}/E$ at the end of training is smaller for larger learning rates (See Appendix A.4 for details).

large learning rates, gradient descent ricochets from peak to peak, until it either diverges or lands in the direct vicinity of a minimum, which is sensitively dependent on initialization (Figure A.1).

To go beyond the limits of implicit gradient regularization, we can explicitly regularize this model using Equation 8 to obtain a regularized loss $E_\mu(a,b) = E(a,b) + \mu \left(|a|^2 + |b|^2\right) x^2 E(a,b)$. Now, if we numerically integrate $\dot{a} = -\nabla_a E_\mu(a,b)$ and $\dot{b} = -\nabla_b E_\mu(a,b)$ starting from the same initial point $(a_0, b_0)$, using a very large explicit regularization rate $\mu$ (and using gradient descent with a very small learning rate $h$ for numerical integration, see Appendix A.4) we find that this flow leads to global minima with a small L2 norm (Figure 1a) in the flattest region of the loss surface. This is not possible with IGR, since it would require learning rates so large that gradient descent would diverge.

## 6 IGR AND EGR IN DEEP NEURAL NETWORKS

Next, we empirically investigate implicit gradient regularization and explicit gradient regularization in deep neural networks. We consider a selection of MLPs trained to classify MNIST digits and we also investigate Resnet-18 trained to classify CIFAR-10 images. All our models are implemented using Haiku (Hennigan et al., 2020).

To begin, we measure the size of implicit regularization in MLPs trained to classify MNIST digits with a variety of different learning rates and network sizes (Figure 2). Specifically, we train 5-layer MLPs with $n_l$ units per layer, where $n_l \in \{50, 100, 200, 400, 800, 1600\}$, $h \in \{0.5, 0.1, 0.05, 0.01, 0.005, 0.001, 0.0005\}$, using ReLu activation functions and a cross entropy loss (see Appendix A.5 for further details and see Figures A.3, A.4 and A.5 for training and test data curves). We report $R_{IG}$ and test accuracy at the time of maximum test accuracy for each network that fits the training data exactly. We choose this time point for comparison because it is important for practical applications. We find that $R_{IG}$ is smaller for larger learning rates and larger networks (Figure 2a), consistent with Theorem 3.1 and Prediction 2.1. Next, we measure the loss surface slope in 5-layer MLPs, with 400 units per layer, trained to classify MNIST digits with a range of different learning rates. We find that neural networks with larger learning rates, and hence, with stronger IGR have smaller slopes at the time of maximum test accuracy (Figure 3a). We also measure the loss surface slope in the vicinity of these optima. To do this, we add multiplicative Gaussian

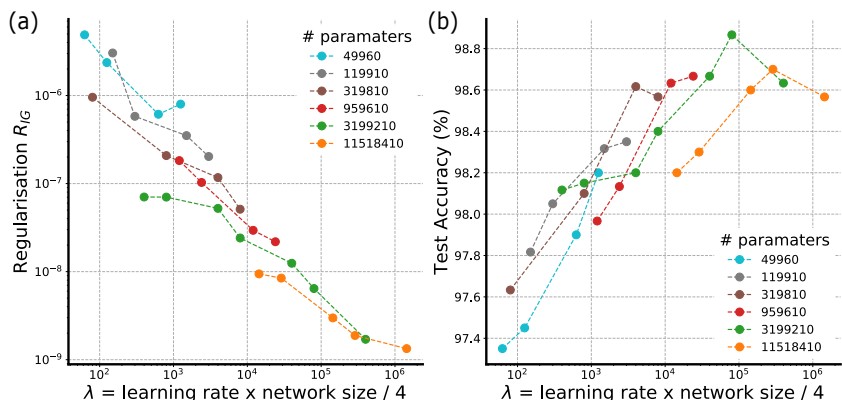

Figure 2: Implicit regularization and test accuracy: (a) Here, each dot represents a different MLP, with different learning rates and network sizes. Implicit gradient regularization $R_{IG}$ is reported for each model, at the time of maximum MNIST test accuracy and $100\%$ train accuracy. We see that models with larger implicit regularization rate $\lambda$ have smaller values of $R_{IG}$. (b) Networks with higher values of $\lambda$ also have higher maximum test accuracy values.

noise to every parameter according to $\theta_p = \theta(1 + \eta)$, where $\theta$ are the parameters of a fully trained model and $\theta_p$ are the parameters after the addition of noise, where $\eta \sim \mathcal{N}(0, \sigma)$. We find that neural networks trained with larger learning rates have flatter slopes and these slopes remain small following larger perturbations (Figure 3a). These numerical results are consistent with our prediction that IGR encourages the discovery of flatter optima (Prediction 2.2)

Next, we observe that improvements in test set accuracy are correlated with increases in regularization rate (Figure 2b), and also with increases in learning rate and network size (Figure A.6). This is consistent with Prediction 2.3. Furthermore, the correlation between test set accuracy and network size $m$ supports our use of network size scaling in Equation 3 and 4.

Next, we explore the robustness of deep neural networks in response to parameter perturbations. In previous work, it has been reported that deep neural networks are robust to a substantial amount of parameter noise, and that this robustness is stronger in networks with higher test accuracy (Morcos et al., 2018). We measure the degradation in classification accuracy as we increase the amount of multiplicative Gaussian noise and find that neural networks with larger learning rates, and hence, with stronger IGR, are more robust to parameter perturbations after training (Figure 3c), consistent with Prediction 2.4. This may explain the origin, in part, of deep neural network robustness.

We also explore IGR in several other settings. For ResNet-18 models trained on CIFAR-10, we find that $R_{IG}$ is smaller and test accuracy is higher for larger learning rates (at the time of maximum test accuracy) (Figure A.7, A.8), consistent again with Theorem 3.1 and Predictions 2.1 and 2.3. We also explore IGR using different stopping time criteria (other than the time of maximum test accuracy), such as fixed iteration time (Figures A.3, A.4), and fixed physical time (Figure A.5) (where iteration time is rescaled by the learning rate, see Appendix A.5 for further information). We explore IGR for full batch gradient descent and for stochastic gradient descent (SGD) with a variety of different batch sizes (Figure A.6) and in all these cases, our numerical experiments are consistent with Theorem 3.1. These supplementary experiments are designed to control for the presence, and absence, of other sources of implicit regularisation - such as model architecture choice, SGD stochasticity and the choice of stopping time criteria.

Finally, we provide an initial demonstration of explicit gradient regularization (EGR). Specifically, we train a ResNet-18 using our explicit gradient regularizer (Equation 8) and we observe that EGR produces a boost of more than 12% in test accuracy (see Figure 3c). This initial experiment indicates that EGR may be a useful tool for training of neural networks, in some situations, especially where IGR cannot be increased with larger learning rates, which happens, for instance, when learning rates are so large that gradient descent diverges. However, EGR is not the primary focus of our work here, but for IGR, which is our primary focus, this experiment provides further evidence that IGR may play an important role as a regularizer in deep learning.

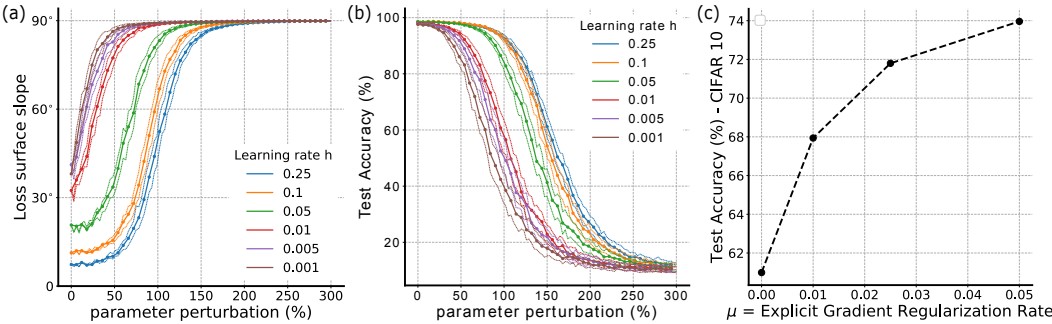

Figure 3: (a) We measure the loss surface slope at the time of maximum test accuracy for models trained on MNIST, and we observe that the loss surface slope is smaller for larger learning rates, and the loss surface slopes remain small for larger parameter perturbations compared to models trained with small learning rates. We perturb our networks by adding multiplicative Gaussian noise to each parameter (up to 300% of the original parameter size). (b) We measure the test accuracy robustness of models trained to classify MNIST digits and see that the robustness increases as the learning rate increases. (Here, solid lines show averages across perturbations and dashed lines demarcate one standard deviation across 100 realizations.) (c) Explicit gradient regularization (EGR) for a ResNet-18 trained on CIFAR-10.

## 7 RELATED WORK

**Implicit regularization:** Many different sources of implicit regularization have been identified, including **early-stopping** (Hardt et al., 2016), **model initialization** (Glorot & Bengio, 2010; Li & Liang, 2018; Nagarajan & Kolter, 2018; Gunasekar et al., 2018; Zhang et al., 2019; Zou et al., 2019), **model architecture** (Li et al., 2018; Lin & Tegmark, 2016; Ma et al., 2020), **stochasticity** (Keskar et al., 2017; Soudry et al., 2018; Roberts, 2018; Ali et al., 2020; Chaudhari & Soatto, 2018; De & Smith, 2020; Mandt et al., 2017; Park et al., 2019; Sagun et al., 2017; Smith & Le, 2018; Wilson et al., 2017; Jastrzebski et al., 2021), **implicit L2 regularity** (Soudry et al., 2018; Neyshabur et al., 2015; Ali et al., 2019; Ji & Telgarsky, 2019; Nacson et al., 2019; Poggio et al., 2019; Suggala et al., 2018), **low rank biases** (Arora et al., 2019a; Gunasekar et al., 2017; Razin & Cohen, 2020) among other possibilities. A number of studies have investigated implicit regularization in the discrete steps of gradient descent for specific datasets, losses, or architectures (Soudry et al., 2018; Neyshabur et al., 2015; Gidel et al., 2019). IGR might also be useful for understanding implicit regularization in deep matrix factorization with gradient descent (Arora et al., 2019a; Gunasekar et al., 2017; Razin & Cohen, 2020), where gradient descent seems to have a low-rank bias. Our work may also provide a useful perspective on the break-even point in deep learning (Jastrzebski et al., 2021). At a break-even point our backward analysis suggests that gradient descent with large learning rates will move toward flatter regions, consistent with this work. Stochastic effects are also likely to contribute to the trajectory at break-even points.

**Learning rate schedules and regimes:** Implicit gradient regularization can be used to understand the role of learning schedules, since learning rate controls the relative strength of implicit regularization and loss optimization. For example, for a cyclical learning rate schedule (Smith, 2017), cyclically varying learning rates between large and small learning rates can be interpreted as a cyclical variation between large and small amounts of IGR (i.e., alternate phases of optimization and regularization). A number of studies have identified various learning rates regimes characterized by different convergence and generalization properties. For instance Li et al. (2019) identifies a small learning rate regime where a network tends to memorize and a large learning rate regime characterized by increased generalization power. This is consistent with IGR which we believe is most useful at the start of training, orienting the search toward flatter regions, and less important in later stages of the training, when a flatter region has been reached, and where convergence to any of the flatter minima is more important. This is also consistent with Jastrzebski et al. (2021) who showed the importance of large learning rates at the beginning of training in encouraging more favourable optimization trajectories. Also Lewkowycz et al. (2020) identifies a lazy phase, a catapult phase, and a divergent phase, which may be related to the range of backward error analysis applicability.

**Neural Tangent Kernel:** The Neural Tangent Kernel (NTK) is especially interesting (Arora et al., 2019b;c; Chizat & Bach, 2019; Jacot et al., 2018; Lee et al., 2019; Oymak & Soltanolkotabi, 2019; Woodworth et al., 2020; Cao & Gu, 2019) since, in the case of the least square loss, the IGR term $R_{IG}$ can be related to the NTK (see Appendix A.3). This is particularly interesting because it suggests that the NTK may play a role beyond the *kernel regime*, into the *rich regime*. In this context, IGR is also related to the trace of the Fisher Information Matrix (Karakida et al., 2019).

**Runge-Kutta methods:** To the best of our knowledge, backward analysis has not been used previously to investigate implicit regularization in gradient based optimizers. However, Runge-Kutta methods have been used to understand old (and devise new) gradient-based optimization methods (Betancourt et al., 2018; Scieur et al., 2017; Zhang et al., 2018; França et al., 2020). A stochastic version of the modified equation was used (Li et al., 2017; Feng et al., 2020) to study stochastic gradient descent in the context of stochastic differential equations and diffusion equations with a focus on convergence and adaptive learning, and very recently França et al. (2020) used backward analysis to devise new optimizers to control convergence and stability.

## 8 DISCUSSION

Following our backward error analysis, we now understand gradient descent as an algorithm that effectively optimizes a modified loss with an implicit regularization term arising through the discrete nature of gradient descent. This leads to several predictions that we confirm experimentally: (i) IGR penalizes the second moment of the loss gradients (Prediction 2.1), and consequently, (ii) it penalizes minima in the vicinity of large gradients and encourages flat broad minima in the vicinity of small gradients (Prediction 2.2); (iii) these broad minima are known to have low test errors, and consistent with this, we find that IGR produces minima with low test error (Prediction 2.3); (iv) the strength of regularization is proportional to the learning rate and network size (Equation 3), (v) consequently, networks with small learning rates or fewer parameters or both will have less IGR and worse test error, and (vi) solutions with high IGR are more robust to parameter perturbations (Prediction 2.4).

It can be difficult to study implicit regularization experimentally because it is not always possible to control the impact of various alternative sources of implicit regularization. Our analytic approach to the study of implicit regularization in gradient descent allows us to identify the properties of implicit gradient regularization independent of other sources of implicit regularization. In our experimental work, we take great care to choose models and datasets that were sufficiently simple to allow us to clearly expose implicit gradient regularization, yet, sufficiently expressive to provide insight into larger, less tractable settings. For many state-of-the-art deep neural networks trained on large real-world datasets, IGR is likely to be just one component of a more complex recipe of implicit and explicit regularization. However, given that many of the favourable properties of deep neural networks such as low test error capabilities and parameter robustness are consistent with IGR, it is possible that IGR is an important piece of the regularization recipe.

There are many worthwhile directions for further work. In particular, it would be interesting to use backward error analysis to calculate the modified loss and implicit regularization for other widely used optimizers such as momentum, Adam and RMSprop. It would also be interesting to explore the properties of higher order modified loss corrections. Although this is outside the scope of our work here, we have provided formulae for several higher order terms in the appendix. More generally, we hope that our work demonstrates the utility of combining ideas and methods from backward analysis, geometric numerical integration theory and machine learning and we hope that our contribution supports future work in this direction.

### ACKNOWLEDGMENTS

We would like to thank Samuel Smith, Soham De, Mihaela Rosca, Yan Wu, Chongli Qin, Mélanie Rey, Yee Whye Teh, Sébastien Racaniere, Razvan Pascanu, Daan Wierstra, Ethan Dyer, Aitor Lewkowycz, Guy Gur-Ari, Michael Munn, David Cohen, Alejandro Cabrera and Shakir Mohamed for helpful discussion and feedback. We would like to thank Alex Goldin, Guy Scully, Elspeth White and Patrick Cole for their support. We would also like to thank our families, especially Wendy; Susie, Colm and Fiona for their support, especially during these coronavirus times.

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

# A   APPENDIX

## A.1   BACKWARD ERROR ANALYSIS OF THE EXPLICIT EULER METHOD

In this section, we provide formulae for higher order backward analysis correction terms for the explicit Euler method, including the first order correction which is required to complete the proof of Theorem 3.1.

We start by restating the general problem addressed by backward error analysis. To begin, consider a first order differential equation

$$\dot{\theta} = f(\theta), \tag{A.1}$$

with vector field $f : \mathbb{R}^m \to \mathbb{R}^m$. The explicit Euler method

$$\theta_n = \theta_{n-1} + h f(\theta_n), \tag{A.2}$$

with step size $h$ produces a sequence of discrete steps $\theta_0, \theta_1, \ldots, \theta_n, \ldots$ approximating the solution $\theta(t)$ of Equation A.1 with initial condition $\theta_0$. (In other word, $\theta_n$ approximates $\theta(nh)$ for $n \geq 0$.) However, at each discrete step the Euler method steps off the continuous solution $\theta(t)$ with a one-step error (or local error) $\|\theta_1 - \theta(h)\|$ of order $\mathcal{O}(h^2)$. Backward error analysis was introduced in numeric integration to study the long term error (or global error) $\|\theta_n - \theta(nh)\|$ of the numerical method. More generally, backward error analysis is useful for studying the long term behavior of a discrete numeric method using continuous flows, such as its numerical phase portrait near equilibrium points, its asymptotic stable orbits, and conserved quantities among other properties (see Hairer & Lubich (1997); Hairer et al. (2006) for a detailed exposition). It stems from the work of Wilkinson (1960) in numerical linear algebra where the general idea in the study of a numeric solution via backward error analysis is to understand it as an exact solution for the original problem but with modified data. When applied to numerical integration of ODEs this idea translates into finding a modified vector field $\tilde{f}$ with corrections $f_i$'s to the original vector field $f$ in powers of the step-size

$$\tilde{f}(\theta) = f(\theta) + h f_1(\theta) + h^2 f_2(\theta) + \cdots \tag{A.3}$$

so that the numeric method steps now exactly follow the (formal) solution of the *modified equation*:

$$\dot{\theta} = \tilde{f}(\theta) \tag{A.4}$$

In other words, if $\theta(t)$ is the solution of the modified equation (A.4) and $\theta_n$ is the $n^{th}$ discrete step of the numerical method, now one has:

$$\theta_n = \theta(nh) \tag{A.5}$$

In general, the sum of the corrections $f_i$'s in (A.3) diverges, and the modified vector field $\tilde{f}$ is only a formal power series. For practical purposes, one needs to truncate the modified vector field. If we truncate the modified vector field up to order $n$ (i.e., we discard the higher corrections $f_l$ for $l \geq n + 1$), the one-step error between the numeric method and the solution of the truncated modified equation is now of order $\mathcal{O}(h^{n+1})$. It is also possible to bound the long term error, but in this section we will only formally derive the higher corrections $f_i$ for the explicit Euler method. (We refer the reader to Hairer & Lubich (1997), for instance, for precise bounds on the error for the truncated modified equation.)

To derive the corrections $f_i$'s for the explicit Euler method, it is enough to consider a single step of gradient descent $\theta + h f(\theta)$ and identify the corresponding powers of $h$ in

$$\theta + h f(\theta) = \text{Taylor}_{|t=0} \, \theta(h)$$

by expanding the solution $\theta(h)$ of the modified equation (A.4) starting at $\theta$ into its Taylor series at zero:

$$\theta(h) = \theta + \sum_{n \geq 1} \frac{h^n}{n!} \theta^{(n)}(0). \tag{A.6}$$

For a gradient flow $f(\theta) = -\nabla E(\theta)$ its Jacobian $f'(\theta)$ is symmetric. In this case, it is natural to look for a modified vector field whose Jacobian is still symmetric (i.e., the higher order corrections can be expressed as gradients). If we assume this, we have the following expression for the higher derivatives of the modified flow solution:

**Lemma A.1.** *In the notation above and with $\tilde{f}$ whose Jacobian is symmetric, we have*

$$\dot{\theta}(0) = \tilde{f}(\theta) \tag{A.7}$$

$$\ddot{\theta}(0) = \frac{d}{d\theta}\frac{\|\tilde{f}(\theta)\|^2}{2} \tag{A.8}$$

$$\theta^{(n)}(0) = \frac{d^{n-2}}{dt^{n-2}}\frac{d}{d\theta}\frac{\|\tilde{f}(\theta)\|^2}{2}, \quad n \geq 2, \tag{A.9}$$

*where $\frac{d^{n-2}}{dt^{n-2}}\frac{d}{d\theta}g(\theta)$ is shorthand to denote the operator $\frac{d^{n-2}}{dt^{n-2}}\big|_{t=0}\frac{d}{d\theta}\big|_{\theta=\theta(t)}g(\theta)$ and $\theta(t)$ is the solution of the modified equation A.4. We will will use this shorthand notation throughout this section.*

*Proof.* By definition $\theta(t)$ is the solution of (A.4) with initial condition $\theta(0) = \theta$, hence $\dot{\theta}(0) = \tilde{f}(\theta)$. Differentiating both sides of (A.4) with respect to $t$, we obtain

$$\ddot{\theta}(t) = \tilde{f}'(\theta(t))\dot{\theta}(t) = \tilde{f}'(\theta(t))\tilde{f}(\theta(t)) = \frac{d}{d\theta}\Big|_{\theta=\theta(t)}\frac{\|\tilde{f}(\theta)\|^2}{2}.$$

The last equation is obtained because we assumed $\tilde{f}'(\theta)$ is symmetric. Now the higher derivatives are obtained by differentiating both sides of the last equation $n - 2$ times with respect to $t$ and setting $t = 0$. □

The previous lemma gives a formula for the derivatives $\theta^{(n)}(0)$. However in order to compare the powers of $h$ we need to expand these derivatives into power of $h$, which is what the next lemma does:

**Lemma A.2.** *In the notation above, we have*

$$\theta^{(n)}(0) = \sum_{k \geq 0} h^k L_{n,k}(\theta) \quad n \geq 2, \tag{A.10}$$

*where we define*

$$L_{n,k}(\theta) = \frac{d^{n-2}}{dt^{n-2}}\frac{d}{d\theta}\sum_{i+j=k}\frac{\langle f_i(\theta), f_j(\theta)\rangle}{2}, \tag{A.11}$$

*with $f_0(\theta)$ being the original vector field $f(\theta)$, and $\langle\,,\,\rangle$ denoting the inner product of two vectors.*

*Proof.* This follows immediately from (A.9) and by expanding $\|\tilde{f}(\theta)\|^2$ in powers of $h$:

$$\frac{\|\tilde{f}(\theta)\|^2}{2} = \frac{1}{2}\langle f_0(\theta) + hf_1(\theta) + \cdots, f_0(\theta) + hf_1(\theta) + \cdots\rangle$$

$$= \sum_{k \geq 0} h^k \sum_{i+j=k}\frac{\langle f_i(\theta), f_j(\theta)\rangle}{2}.$$

□

Putting everything together, we now obtain a Taylor series for the solution of the modified equation as a formal power series in $h$:

**Lemma A.3.** *In the notation above, we have*

$$\theta(h) = \theta + \sum_{l \geq 0} h^{l+1}\big(f_l(\theta) + H_l(f_0, f_1, \ldots, f_{l-1})(\theta)\big), \tag{A.12}$$

*where $f_0$ is the original vector field $f$, $H_0 = 0$ and we define recursively*

$$H_l(f_0, f_1, \ldots, f_{l-1})(\theta) = \sum_{\substack{n+k=l+1 \\ n \geq 2,\, l \geq 0}}\frac{1}{n!}L_{n,k}(\theta) \tag{A.13}$$

*Proof.* Replacing $\theta^{(n)}(0)$ by their expression in (A.10) in the Taylor series (A.6), we obtain

$$
\begin{aligned}
\theta(h) &= \theta + h\tilde{f}(\theta) + \sum_{n\geq 2}\sum_{k\geq 0}\frac{h^{n+k}}{n!}L_{n,k}(\theta) \\
&= \theta + \sum_{l\geq 0}h^{l+1}\big(f_l(\theta) + \sum_{\substack{n+k=l+1 \\ n\geq 2,\, l\geq 0}}\frac{1}{n!}L_{n,k}(\theta)\big),
\end{aligned}
$$

which finishes the proof. $\qquad\square$

Now comparing the Taylor series for the modified equation solution in its last form in (A.12) with one step of the Euler method

$$
\theta + hf(\theta) = \theta + hf(\theta) + \sum_{l\geq 1}h^{l+1}\big(f_l(\theta) + H_l(\theta)\big)
$$

for each order of $h$, we obtain the following proposition:

**Proposition A.4.** *The corrections $f_i$'s for the Euler method modified equation in (A.3) are given by the general recursive formula:*

$$
f_l(\theta) = -\sum_{\substack{n+k=l+1 \\ n\geq 2,\, l\geq 0}}\frac{1}{n!}L_{n,k}(\theta), \tag{A.14}
$$

*where the $L_{n,k}$ are defined by Equation A.11.*

Let us use (A.14) to compute the first order correction in the modified equation for the Euler explicit method:

**Example A.5.** *Suppose $f = -\nabla E$, then the first order correction is*

$$
f_1(\theta) = -\frac{1}{2}L_{2,0}(\theta) = -\frac{1}{4}\frac{d}{d\theta}\|\nabla E(\theta)\|^2
$$

*which is the only correction we use in Theorem 3.1. The remarkable thing is that now the first two terms of (A.3) can be understood as the gradient of a modified function, yielding the following form for the modified equation (A.4):*

$$
\dot{\theta} = -\nabla\tilde{E} + \mathcal{O}(h^2),
$$

*with $\tilde{E} = E + \frac{h}{4}\|\nabla E\|^2$, which is the main result of Theorem 3.1.*

## A.2 GEOMETRY OF IMPLICIT GRADIENT REGULARIZATION

In this section, we provide all the details for a proof of Proposition 3.3 and for our claim concerning the relationship between the loss surface slope and the implicit gradient regularizer, which we package in Corollary A.6.1. The geometry underlying implicit gradient regularization makes it apparent that gradient descent has a bias toward flat minima (Keskar et al., 2017; Hochreiter & Schmidhuber, 1997).

To begin with, consider a loss $E$ over the parameter space $\theta \in \mathbb{R}^m$. The loss surface is defined as the graph of the loss:

$$
S = \{(\theta, E(\theta)) : \theta \in \mathbb{R}^m\} \subset \mathbb{R}^{m+1}.
$$

It is a submanifold of $\mathbb{R}^{m+1}$ of co-dimension 1, which means that the space of directions orthogonal to $S$ at a given point $(\theta, E(\theta))$ is spanned by a single unit vector, the normal vector $N(\theta)$ to $S$ at $(\theta, E(\theta))$. There is a natural parameterization for surfaces given by the graphs of functions, the Monge parameterization, where the local chart is the parameter plane: $\theta \to (\theta, E(\theta))$. Using this parameterization it is easy to see that the tangent space to $S$ at $(\theta, E(\theta))$ is spanned by the tangent vectors:

$$
v_i(\theta) = (0, \ldots, 1, \ldots, 0, \nabla_{\theta_i}E(\theta)),
$$

for $i = 1, \ldots, m$ and where the 1 is at the $i^{th}$ position. Now that we have the tangent vectors, we can verify that the following vector is the normal vector, since its inner product with all the tangent vectors is zero and its norm is one:

$$N(\theta) = \frac{1}{\sqrt{1 + \|\nabla E(\theta)\|^2}} (-\nabla_{\theta_1} E(\theta), \ldots, -\nabla_{\theta_m} E(\theta), 1)$$

We can compute the cosine of the angle between the normal vector $N(\theta)$ at $(\theta, E(\theta))$ and the vector $\hat{z} = (0, \ldots, 0, 1)$ that is perpendicular to the parameter plane by taking the inner product between these two vectors, immediately yielding the following Proposition:

**Proposition A.6.** *Consider a loss $E$ and its loss surface $S$ as above. The cosine of the angle between the normal vector $N(\theta)$ to the loss surface at $(\theta, E(\theta))$ and the vector $\hat{z} = (0, \ldots, 0, 1)$ perpendicular to the parameter plane can be expressed in terms of the implicit gradient regularizer as follows:*

$$\langle N(\theta), \hat{z} \rangle = \frac{1}{\sqrt{1 + mR_{IG}(\theta)}} \tag{A.15}$$

Now observe that if $\langle N(\theta), \hat{z} \rangle$ is zero this means that the tangent plane to $S$ at $(\theta, E(\theta))$ is orthogonal to the parameter space, in which case the loss surface slope is maximal and infinite at this point! On the contrary, when $\langle N(\theta), \hat{z} \rangle$ is equal to 1, the tangent plane at $(\theta, E(\theta))$ is parallel to the parameter plane, making $S$ look like a plateau in a neighborhood of this point.

Let us make precise what we mean by loss surface slope. First notice that the angle between $\hat{z}$ (which is the unit vector normal to the parameter plane) and $N(\theta)$ (which is the vector normal to the tangent plane to $S$) coincides with the angle between this two planes. We denote by $\alpha(\theta)$ this angle:

$$\alpha(\theta) = \arccos\langle N(\theta), \hat{z} \rangle. \tag{A.16}$$

Now, we define the *loss surface slope* at $(\theta, E(\theta))$ by the usual formula

$$\text{slope}(\theta) = \tan \alpha(\theta). \tag{A.17}$$

As we expect, when the loss surface slope is zero this means that the tangent plane to the loss surface is parallel to the parameter plane (i.e., $\alpha(\theta) = 0$), while when the slope goes to infinity it means the tangent plane is orthogonal to the parameter plane (i.e., $\alpha(\theta) = \pi/2$).

The following corollary makes it clear that implicit gradient regularization in gradient descent orients the parameter search for minima toward flatter regions of the parameter space, or flat minima, which have been found to be more robust and to possess more generalization power (see Keskar et al. (2017); Hochreiter & Schmidhuber (1997)):

**Corollary A.6.1.** *The slope of the loss surface $S$ at $(\theta, E(\theta))$ can be expressed in terms of the implicit gradient regularizer as follows:*

$$\text{slope}(\theta) = \sqrt{mR_{IG}(\theta)} \tag{A.18}$$

*Proof.* From (A.15) and the fact that $\cos \alpha(\theta) = \langle N(\theta), \hat{z} \rangle$, we have that

$$\frac{1}{\cos^2 \alpha(\theta)} = 1 + mR_{IG}(\theta).$$

Now basic trigonometry tells us that in general $1/\cos^2 \alpha = 1 + \tan^2 \alpha$, which implies here that $\tan^2 \alpha(\theta) = mR_{IG}(\theta)$. Taking the square root of this last expression finishes the proof. $\square$

**Remark A.7.** *Corollary A.6.1 gives us a very clear understanding of implicit gradient regularization. Namely, the quantity that is regularized is nothing other than the square of the slope $R_{IG}(\theta) = \frac{1}{m} \text{slope}^2(\theta)$ and the modified loss becomes $\tilde{E}(\theta) = E(\theta) + \frac{h}{4} \text{slope}^2(\theta)$. For* explicit *gradient regularization, we can now also understand the explicitly regularized loss in terms of the slope:*

$$E_\mu(\theta) = E(\theta) + \mu \, \text{slope}^2(\theta),$$

*This makes it clear that this explicit regularization drives the model toward flat minima (with zero slope).*

**Remark A.8.** *There is another connection between IGR and the underlying geometry of the loss surface through the metric tensor. It is a well-known fact from Riemannian geometry that the metric tensor $g(\theta)$ for surfaces in the Monge parameterization $\theta \to (\theta, E(\theta))$ has the following form:*

$$g_{ij}(\theta) = \delta_{ij} + \nabla_{\theta_i} E(\theta) \nabla_{\theta_j} E(\theta),$$

*where $\delta_{ij}$ is the Kronecker delta. Now the determinant $|g|$, which defines the local infinitesimal volume element on the loss surface, can also be expressed in terms of the implicit gradient regularizer: Namely, $|g(\theta)| = 1 + \|\nabla E(\theta)\|^2 = 1 + mR_{IG}(\theta)$. Solving this equation above for $R_{IG}$, we obtain a geometric definition for the implicit gradient regularizer:*

$$R_{IG}(\theta) = \frac{1}{m}(|g(\theta)| - 1), \tag{A.19}$$

*which incidentally is zero when the surface looks like an Euclidean space.*

We conclude this section by showing that the increase in parameter norm can be bounded by the loss surface slope at each gradient descent step.

**Proposition A.9.** *Let $\theta_n$ be the parameter vector after $n$ gradient descent updates. Then the increase in parameter norm is controlled by the loss surface slope as follows:*

$$\left| \|\theta_{n+1}\| - \|\theta_n\| \right| \leq h \,\mathrm{slope}(\theta_n) \tag{A.20}$$

*Proof.* The triangle inequality applied to one step of gradient descent $\|\theta_{n+1}\| = \|\theta_n - h\nabla E(\theta_n)\|$ yields $(\|\theta_{n+1}\| - \|\theta_n\|) \leq h\|\nabla E(\theta_n)\|$, which concludes the proof, since the gradient norm coincides with the loss surface slope. □

We now prove a proposition that relates the geometry of the minima of $E$ and $\tilde{E}$.

**Proposition A.10.** *Let $E$ be a non-negative loss. Then local minima of $E$ are local minima of the modified loss $\widetilde{E}$. Moreover, the two losses have the same interpolating solutions (i.e., locus of zeros).*

*Proof.* Since $\nabla \widetilde{E} = \nabla E + \frac{h}{2} D^2 E \nabla E$, where $D^2 E$ is the Hessian of $E$, it is clear that a critical point of $E$ is a critical point of $\widetilde{E}$. Suppose now that $\theta^*$ is a local minimum of $E$. This means that there is a neighbourhood of $\theta^*$ where $E(\theta) \geq E(\theta^*)$. We can add $\frac{h}{4}\|\nabla E(\theta)\|^2$ on the left of this inequality, since it is a positive quantity, and we can also add $\frac{h}{4}\|\nabla E(\theta^*)\|^2$ on the right of the inequality, since it is zero. This shows that in a neighborhood of $\theta^*$, we also have that $\widetilde{E}(\theta) \geq \widetilde{E}(\theta^*)$. This means that $\theta^*$ is also a local minimum of $\widetilde{E}$. Finally, let us see that $E$ and $\widetilde{E}$ share the same locus of zeros. Let $\theta$ be a zero of $E$. Since $E$ is non-negative and $E(\theta) = 0$ then $\theta$ is a global minima, which implies that $\nabla E(\theta) = 0$ also, and hence $\widetilde{E}(\theta) = 0$. Now for positive $E$, $\widetilde{E}(\theta) = 0$ trivially implies $E(\theta) = 0$. □

## A.3 THE NTK CONNECTION

In the case of the least square loss, the modified equation as well as the implicit gradient regularizer take a very particular form, involving the Neural Tangent Kernel (NTK) introduced in Jacot et al. (2018).

**Proposition A.11.** *Consider a model $f_\theta : \mathbb{R}^d \to \mathbb{R}^c$ with parameters $\theta \in \mathbb{R}^m$ and with least square loss $E(\theta) = \sum_{i=1}^n \|f_\theta(x_i) - y_i\|^2$. The modified loss can then be expressed as*

$$\tilde{E}(\theta) = E(\theta) + h \sum_{i,j=1}^n \epsilon_i^T(\theta) K_\theta(x_i, x_j) \epsilon_j(\theta), \tag{A.21}$$

*where $K_\theta$ is the Neural Tangent Kernel defined by*

$$K_\theta(x_i, x_j) := \nabla \epsilon_i(\theta)^T \nabla \epsilon_j(\theta), \tag{A.22}$$

*where $\epsilon_k(\theta) = f_\theta(x_k) - y_k \in \mathbb{R}^c$ is the error vector on data point $x_k$. In the particular case when the model output is one-dimensional, we can write the modified loss compactly as follows:*

$$\tilde{E}(\theta) = \epsilon(\theta)^T (1 + K_\theta) \epsilon(\theta), \tag{A.23}$$

*Proof.* Let us begin by computing the gradient

$$
\begin{aligned}
\nabla E(\theta) &= \sum_{i=1}^{n} \nabla \|f_\theta(x) - y\|^2 \\
&= 2 \sum_{i=1}^{n} \langle \nabla_\theta f_\theta(x), (f_\theta(x) - y) \rangle \\
&= 2 \sum_{i=1}^{n} \nabla \epsilon_i(\theta) \epsilon_i(\theta),
\end{aligned}
$$

since $\nabla \epsilon_i(\theta) = \nabla_\theta f_\theta(x_i)$ is a matrix. Using that result, we can compute the implicit gradient regularizer in this case:

$$
\begin{aligned}
R_{IG}(\theta) &= \frac{1}{m} \langle \nabla E(\theta), \nabla E(\theta) \rangle \\
&= \frac{4}{m} \sum_{i,j=1}^{n} \langle \nabla \epsilon_i(\theta) \epsilon_i(\theta), \nabla \epsilon_j(\theta) \epsilon_j(\theta) \rangle \\
&= \frac{4}{m} \sum_{i,j=1}^{n} \epsilon_i(\theta)^T \nabla \epsilon_i(\theta)^T \nabla \epsilon_j(\theta) \epsilon_j(\theta) \\
&= \frac{4}{m} \sum_{i,j=1}^{n} \epsilon_i(\theta)^T K_\theta(x_i, x_j) \epsilon_j(\theta),
\end{aligned}
$$

which concludes the first part of the proof. Now when the model output is one-dimensional, then the $\epsilon_i(\theta)$ are no longer vectors but scalars. We can then collect them into a single vector $\epsilon(\theta) = (\epsilon_1(\theta), \ldots, \epsilon_n(\theta))$. Similarly, the terms $K_\theta(x_i, x_j)$ are no longer matrices but scalars, allowing us to collect them into a single matrix $K_\theta$. In this notation we now see that the original loss can be written as $E = \epsilon^T \epsilon$, yielding $\tilde{E} = \epsilon^T (1 + K_\theta) \epsilon$ for the modified loss, concluding the proof.

$\square$

For the least square loss, we see that the IGR is expressed in terms of the NTK. Therefore the NTK entries will tend to be minimized during gradient descent. In particular, the cross terms $K_\theta(x_i, x_j)$ will be pushed to zero. This means that gradient descent will push the maximum error direction $\nabla \epsilon_k(\theta)$ at different data points to be orthogonal to each other (i.e., $\nabla \epsilon_k(\theta)^T \nabla \epsilon_l(\theta) \simeq 0$). This is good, since a gradient update is nothing other than a weighted sum of these error directions. If they are orthogonal, this means that the gradient update contribution at point $x_k$ will not affect the gradient update contribution at point $x_l$, so the individual data point corrections are less likely to nullifying each other as gradient descent progresses.

## A.4 EXPERIMENT DETAILS FOR THE 2-D LINEAR MODEL

In this section, we provide supplementary results (Figure A.1), hyper-parameter values (Table A.1) and modified loss derivations for the two parameter model described in Section 5. This model has a loss given by:

$$
E = (y - abx)^2 / 2 \tag{A.24}
$$

where $a, b \in \mathbb{R}$ are the model parameters and $x, y \in \mathbb{R}$ is the training data.

The implicit regularization term for this model can be calculated using Equation 4, yielding:

$$
R_{IG} = \frac{(\nabla_a E)^2 + (\nabla_b E)^2}{2} = (a^2 + b^2) x^2 E. \tag{A.25}
$$

The implicit regularization rate can be calculated using Equation 3, yielding:

$$
\lambda = mh/4 = h/2. \tag{A.26}
$$

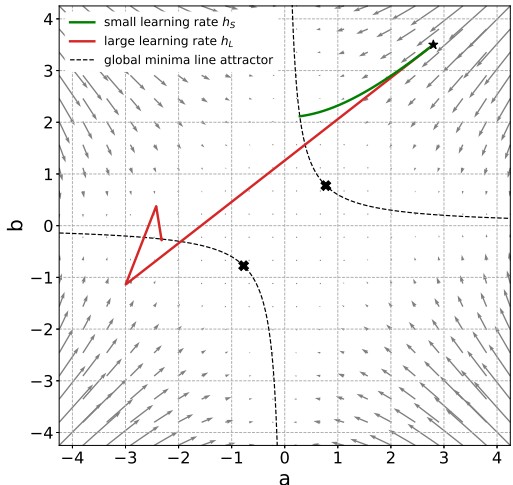

Figure A.1: Gradient descent with large learning rates: In this phase space plot, the same loss surface as in Figure 1 is represented, but showing a larger area of the phase space. We plot two gradient descent trajectories, originating from the same initial point ($a_0 = 2.8$, $b_0 = 3.5$). One has a small learning rate $h_S = 10^{-3}$ and the other has a much larger learning rate $h_L = 1.8 \times 10^{-1}$. The gradient descent trajectory with larger learning rate ricochets across the loss surface, stepping over line attractors until it happens to land in a low gradient region of the loss surface where it converges toward a global minima.

The modified loss can be calculated using Equation 2, yielding:

$$\widetilde{E} = E + \lambda R_{IG} = E\left(1 + \lambda\left(a^2 + b^2\right)x^2\right). \tag{A.27}$$

Here, we can see that the global minima for $E(a, b)$ (i.e. the zeros) are the same as the global minima for $\widetilde{E}(a, b)$ since $1 + \lambda\left(a^2 + b^2\right)x^2$ is positive. However, as we will see, the corresponding gradient flows are different.

The exact modified gradient flow for the modified loss is given by:

$$
\begin{aligned}
\dot{a} &= -\nabla_a\widetilde{E} = -\nabla_a E\left(1 + \lambda\left(a^2 + b^2\right)x^2\right) - \lambda\left(2ax^2\right)E \\
\dot{b} &= -\nabla_b\widetilde{E} = -\nabla_b E\left(1 + \lambda\left(a^2 + b^2\right)x^2\right) - \lambda\left(2bx^2\right)E,
\end{aligned} \tag{A.28}
$$

The exact gradient flow for the original loss is given by

$$
\begin{aligned}
\dot{a} &= -\nabla_a E \\
\dot{b} &= -\nabla_b E,
\end{aligned} \tag{A.29}
$$

The exact numerical flow of gradient descent is given by

$$
\begin{aligned}
a_{n+1} &= a_n - h\nabla_a E \\
b_{n+1} &= b_n - h\nabla_b E,
\end{aligned} \tag{A.30}
$$

Table A.1: Experiment hyper-parameters for the two-parameter model

|  | Initial point I | Initial point II |
|---|---|---|
| $(a_0, b_0)$ | $(2.8, 3.5)$ | $(75, 74.925)$ |
| $(x, y)$ | $(1, 0.6)$ | $(1, 0.6)$ |
| $h_S$ | $10^{-3}$ | $10^{-6}$ |
| $h_M$ | $2.5 \times 10^{-2}$ | $0.5 \times 10^{-4}$ |
| $h_L$ | $1.8 \times 10^{-1}$ | n/a |
| $\mu$ | $0.5$ | $4.1 \times 10^{-2}$ |
| $h_{Euler}$ | $10^{-4}$ | $5 \times 10^{-7}$ |

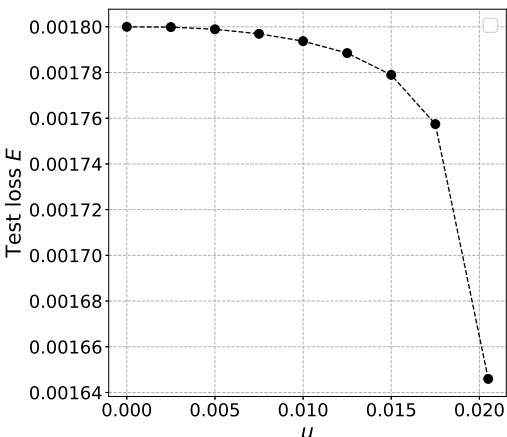

Figure A.2: Test error for a 2-d linear model: We evaluate test error using a test data point given by $(x = 0.5, y = 3/5)$ for a model trained on a training data point given by $(x = 1, y = 3/5)$. We see that models with larger amounts of EGR have smaller test errors, consistent with Prediction 2.3. This happens because gradient regularisation leads to flatter solutions, where the difference between train and test data incurs a smaller cost.

where $(a_n, b_n)$ are the parameters at iteration step $n$. For this model, we have $\nabla_a E = -bx(y - abx)$ and $\nabla_b E = -ax(y - abx)$.

**Remark A.12.** *In vector notation, we see that the modified gradient flow equation is*

$$\dot{\theta} = -(1 + \lambda x^2 \|\theta\|^2)\nabla E(\theta) - (2\lambda x^2 E)\theta,$$

*with $\theta = (a, b)$. The last term $-(2\lambda x^2 E)\theta$ is a central vector field re-orienting the original vector field $\nabla E$ away from the steepest slopes and toward the origin which coincides in this example with flatter regions where the minimal norm global minima are located. This phenomenon becomes stronger for parameters further away from the origin, where, coincidentally the slopes are the steepest. Specifically,* $\text{slope}(\theta) = \|\theta\| |x| \sqrt{2E}$.

In Figure 1a, we plot the trajectories of four different flows starting from the same initial point $(a_0, b_0)$, to illustrate the impact of IGR. We look at two initial points, chosen to illustrate the behaviour of gradient descent in different settings. The full set of hyper-parameters used for this experiment is given in Table A.1. First, we calculate the numerical flow of gradient descent with a small learning rate $h_S$ using Equation A.30. Next, we plot the numerical flow of gradient descent with a moderate learning rate $h_M$ using Equation A.30. We then calculate the modified gradient flow by solving Equation A.28 numerically using the Euler method, starting from initial point $(a_0, b_0)$ and using $\lambda = h_M/2$. For this numerical calculation, we use a very small Euler method step size $h_{Euler}$ so that the Euler method follows the gradient flow of the modified loss accurately. We observe that this modified flow is close to the numerical flow of gradient descent, consistent with Theorem 3.1.

We also plot the trajectory of gradient descent for a large learning rate $h_L$, where backward analysis is no longer applicable (Fig. A.1) and observe that gradient descent ricochets across the loss surface, stepping over line attractors until it lands in a low gradient region of the loss surface where it converges toward a global minima. This large learning rate regime can be unstable. For larger learning rates, or for different initial positions, we observe that gradient descent can diverge in this regime.

In Figure 1b, we explore the asymptotic behaviour of gradient descent by measuring $R/E$ after convergence for a range of models, all initialized at $a_0 = 2.8$ and $b_0 = 3.5$, with a range of learning rates.

We also explore the impact of explicit gradient regularization, using Equation 8 to define the explicitly regularized modified loss for our two-parameter model:

$$E_\mu = E\left(1 + \mu\left(a^2 + b^2\right)x^2\right). \tag{A.31}$$

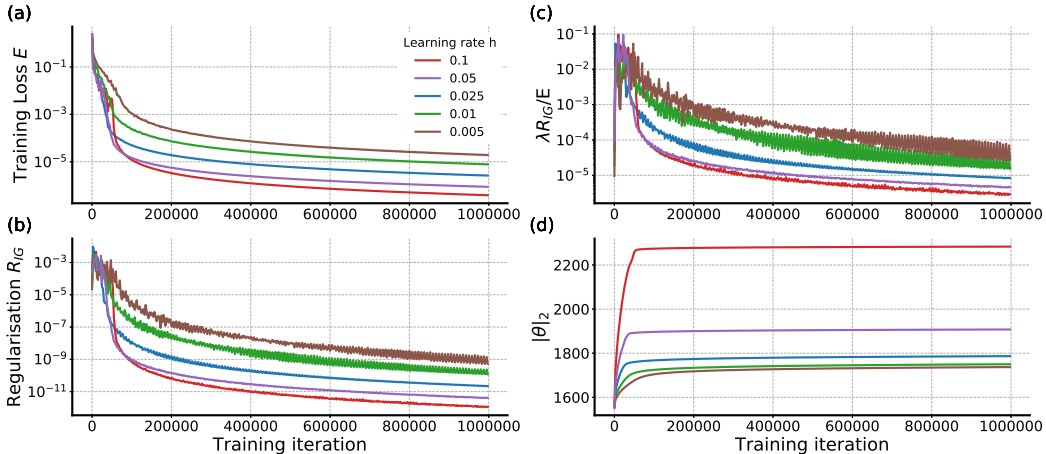

Figure A.3: Implicit gradient regularization in a deep neural network: For an MLP trained to classify MNIST digits, we see that (a) the training loss $E$ and (b) the regularization value $R_{IG}$ both decrease during training, consistent with backward error analysis (Equation 2). (c) The ratio of $\lambda R_{IG}/E$ is smaller for networks that have larger learning rates, consistent with the prediction that the learning rate controls the regularization rate (Equation 3). (d) The parameter magnitudes grow during training, and this growth decreases as $R_{IG}$ decreases and finally plateaus before the training loss does, consistent with Proposition A.9.

We use this modified loss for gradient descent:

$$
\begin{aligned}
a_{n+1} &= a_n - h_{Euler}\nabla_a E_\mu \\
b_{n+1} &= b_n - h_{Euler}\nabla_b E_\mu,
\end{aligned}
\tag{A.32}
$$

Here, we have used a very small learning rate, $h_{Euler}$ (Table A.1) and a very large value of $\mu$ (Table A.1). This allows us to achieve stronger regularization, since $\mu$ can be increased to a large value where gradient descent with $h = 2\mu$ would diverge. We observe that EGR can decrease the size of $R_{IG}$ after training (Figure 1a) and can increase test accuracy (Figure A.2).

## A.5 DEEP NEURAL NETWORK EXPERIMENT DETAILS

In this section, we provide further details for the calculation of $R_{IG}(\theta)$ in a deep neural network (Figure 2, 3, A.3, A.4, A.5, A.6, A.7, A.8). For all these experiments, we use JAX (Bradbury et al., 2018) and Haiku (Hennigan et al., 2020) to automatically differentiate and train deep neural networks for classification. Conveniently, the loss gradients that we compute with automatic differentiation are the same loss gradients that we need for the calculation of $R_{IG}(\theta)$.

We calculate the size of implicit gradient regularization $R_{IG}(\theta)$, during model training, using Equation 4. We observe that $R_{IG}(\theta)$, the loss $E(\theta)$ and the ratio $R_{IG}/E(\theta)$ all decrease as training progresses, for all learning rates considered (Figure A.3). We also observe that the parameter magnitudes grow during training, and this growth slows as $R_{IG}(\theta)$ becomes small, in agreement with Proposition A.9. After a sufficiently large fixed number of training steps, we see that models with larger learning rates have much smaller values of $R_{IG}(\theta)$ relative to $E(\theta)$, which appears to be consistent with Prediction 2.1. However, the speed of learning clearly depends on the learning rate $h$ so it may not be reasonable to compare models after a fixed number of training iterations. Instead of stopping after a fixed number of iterations, we could stop training after $n = T/h$ iterations, where $T$ is the fixed physical time that naturally occurs in our backward analysis (Equation A.5). Again, we find that models with larger learning rates have lower values of $R_{IG}(\theta)$ and $E(\theta)$ after a sufficiently large amount of physical training time $T$ (Figure A.5). However, even for fixed physical time comparisons, we still need to choose an arbitrary physical time point $T$ for making comparisons between models. The choice of stopping time is effectively an unavoidable form of implicit regularization. Instead of fixed iteration time or fixed physical time, we use the time of maximum test accuracy as the stopping time for model comparison in Figure 2, 3 and A.6. We choose this option because it is the most

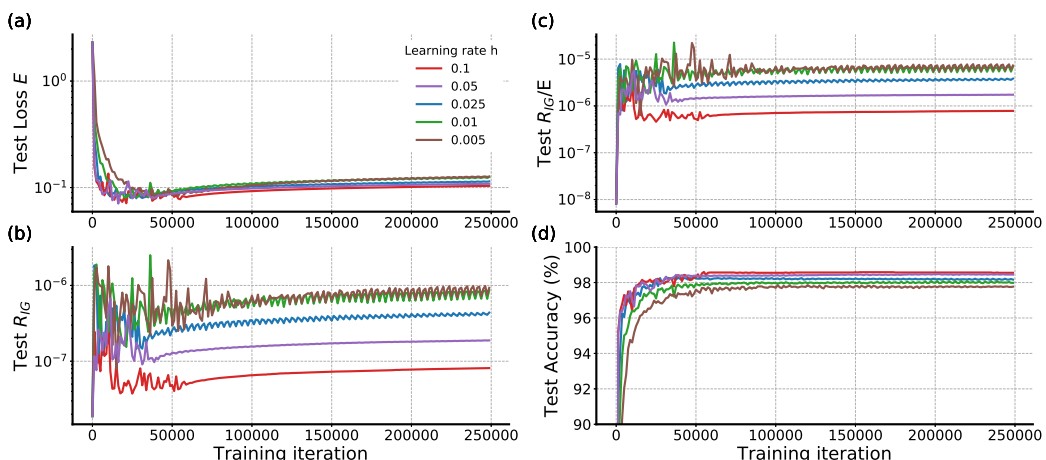

Figure A.4: Implicit gradient regularization with test data: (a) The test loss $E$ and (b) the test regularisation value $R_{IG}$ both decrease initially before increasing again. (c) The ratio of $R_{IG}/E$ is smaller for networks that have larger learning rates, consistent with Equation 2. (d) The test accuracy increases during training, until it reaches a maximum value before starting to decrease again slightly. The models used in this figure are the same as those reported in Figure 2, but using test data instead of training data.

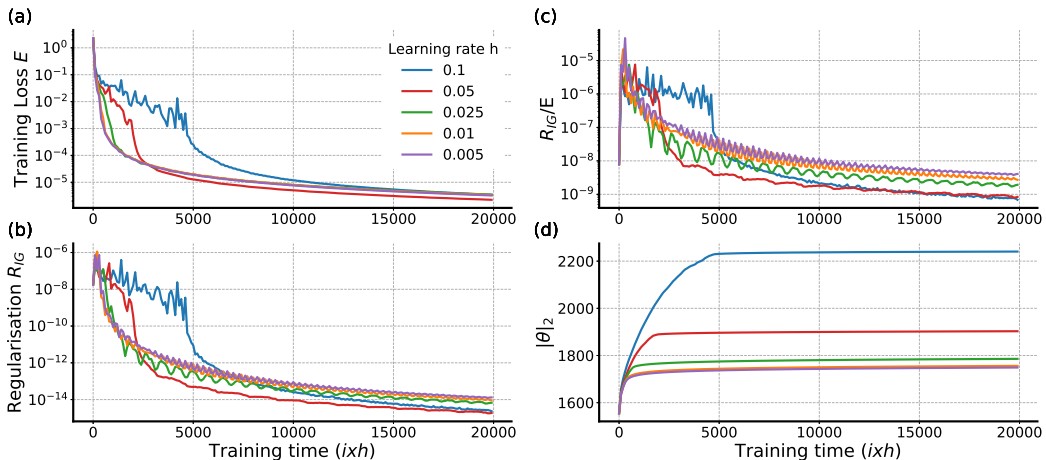

Figure A.5: Implicit gradient regularization for networks trained for equal amounts of physical training time $T = i \times h$, where $i$ is the number of training iterations. Since physical training time depends on learning rate $h$, models with large learning rates are trained for far fewer iterations, and are exposed to far fewer epochs of training data than models with small learning rates. Nonetheless, following a sufficiently long period of physical training time, the ratio $R_{IG}(\theta)/E(\theta)$ is smaller for models with larger learning rates, consistent with Theorem 3.1.

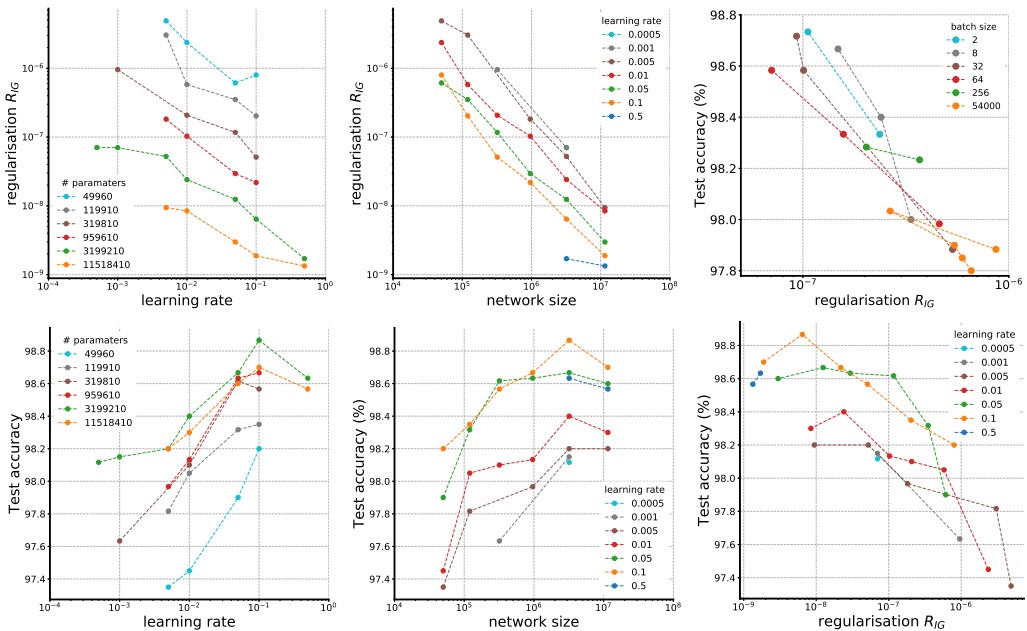

Figure A.6: The relationship between learning rate, network size, $R_{IG}$ and test accuracy, using the same data as in Figure 2. We see that $R_{IG}$ typically decreases as the learning rate increases, or, as the network size increases. We also see that test accuracy typically increases as the learning rate increases, or, as the network size increases. Finally, we observe that test error is strongly correlated with the size of $R_{IG}$ after training.

useful time point for most real-world applications. For each model, we calculate $E(\theta)$, $R_{IG}(\theta)$ and the test accuracy at the time of maximum test accuracy (which will be a different iteration time for each model) (Figure A.4). The observation that (i) fixed iteration stopping time, (ii) fixed physical stopping time, and (iii) maximum test accuracy stopping time all have smaller values of $R_{IG}(\theta)/E(\theta)$ for larger values of $\lambda$, consistent with Prediction 2.1, indicates that the relationships between these quantities cannot be trivially explained to be a consequence of a particular choice stopping time regularization. In these examples, we use $n_l = 400$ (corresponding to $\sim 9.6 \times 10^6$ parameters) with batch size 32.

In Figure 2 and Figure A.6 we report $R_{IG}(\theta)$ and test accuracy at the time of maximum test accuracy for a selection of networks of different size, trained with different learning rates. For models with sufficient capacity to solve the training task and simultaneously minimize $R_{IG}(\theta)$, we expect $R_{IG}(\theta)/E$ and test error to decrease as $\lambda$ increases (Prediction 2.1 and 2.3). To expose this behaviour, we exclude models that fail to reach $100\%$ MNIST training accuracy, such as models that diverge (in the large learning rate regime), and models with excessively small learning rates, which fail to solve the task, even after long training periods. We observe that test error is strongly correlated with the size of $R_{IG}$ after training (Figure A.6). We also confirm this for a range of batch sizes, including full batch gradient descent (Figure A.6, top right) with $n_l = 400$, and for SGD with batch size 32 across a range of learning rates and network sizes (Figure A.6, bottom right).

Finally, to explore IGR and EGR for a larger model we trained a ResNet-18 to classify CIFAR-10 images using Haiku (Hennigan et al., 2020). We used stochastic gradient descent for the training with a batch size of 512 for a range of learning rates $l \in \{0.005, 0.01, 0.05, 0.1, 0.2\}$. We observe the same behaviour as in the MNIST experiments: as the learning rate increases, the values of $R_{IG}$ decrease (Prediction 2.1), the test accuracy increases (Prediction 2.3) and the optimization paths follow shallower slopes leading to broader minima (Prediction 2.2). The experimental results are summarized by the training curves displayed in Figure A.7 and in Figure A.8, where we plot the relation between learning rate, $R_{IG}$, and test accuracy taken at the time of maximum test accuracy for each training curve.

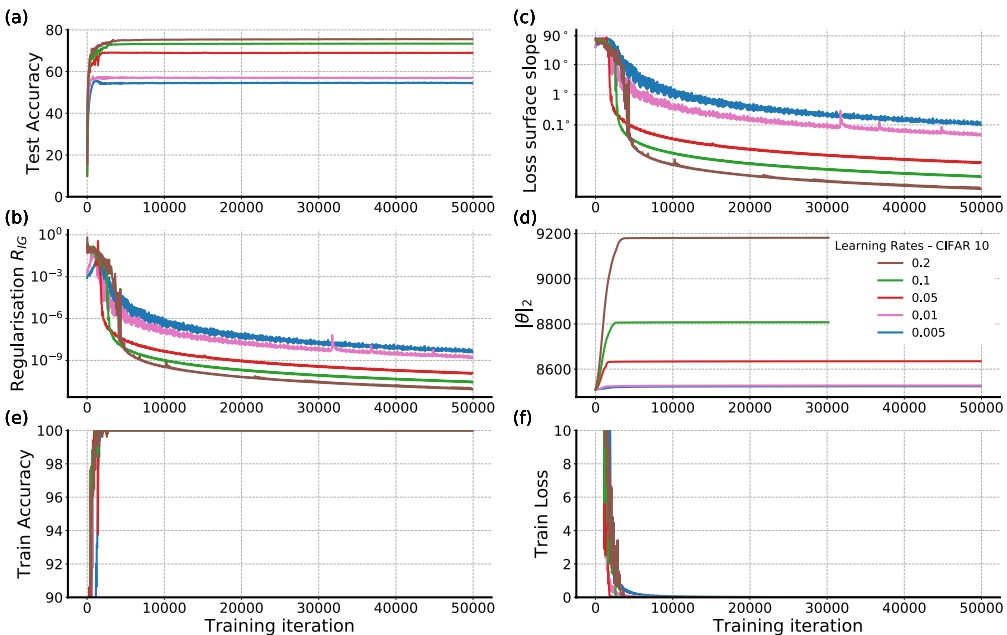

Figure A.7: Implicit gradient regularization for Resnet-18 trained on CIFAR-10: (a) The test accuracy increases and (b) the regularization value $R_{IG}$ decreases as the learning rate increases. (c) The optimization paths follow shallower loss surface slopes as the learning rate increases. (d) We also report the L2 norm of the model parameters. (e) and (f) The interpolation regime is reached after a few thousand iterations.

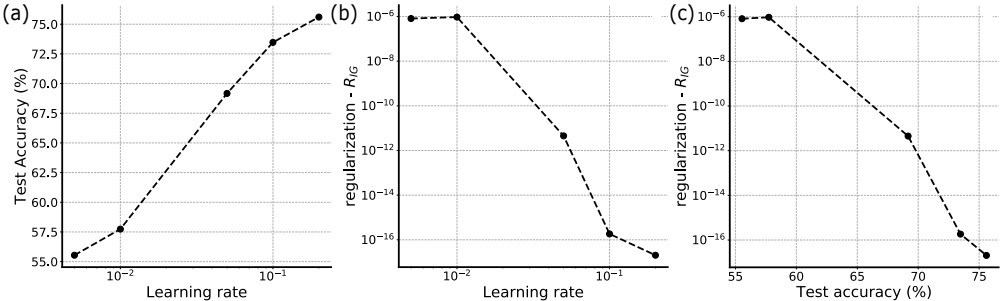

Figure A.8: The relationship between learning rate, $R_{IG}$, and test accuracy, using the same data as in Figure A.7 for various Resnet-18 models trained on CIFAR-10. We see that learning $R_{IG}$ typically decreases and the test accuracy increases as the learning rate increases. Finally, we also observe that test accuracy is strongly correlated with the size of $R_{IG}$. All the values are taken at the time of maximum test accuracy.

