# OpenReview forum: "Implicit Gradient Regularization"
_ICLR.cc/2021/Conference — ICLR 2021 Poster_

### Official Review · AnonReviewer1 · 2020-10-18
**A really nice paper about implicit regularization from the discreteness of gradient descent: the problem is clearly motivated, the authors explain the phenomenon they intend to study, the analysis of the phenomenon is clear, the toy model is nice and clear, and then confirming experiments follow.**

**Rating:** 7
**Confidence:** 5

**Review:**

This paper provides a unique perspective on the implicit regularization effect of gradient descent that has been observed and studied previously. The authors point out that the discrete steps taken by the gradient descent updates means that the path followed through the optimization landscape is not that of steepest descent, but some alternate path. Thinking of GD as trying to solve the continuous time evolution equation implied by GD, they analyze the errors that the actual updates make in solving this equation. Given these errors, they construct an alternate ODE whose solution has a discretization that is precisely the GD updates (up to higher order corrections in the learning rate). Determining the loss implied by this alternative ODE gives an additional term, proportional to the norm squared of the gradient, the learning rate, and the number of parameters. This "Implicit Regularization'' leads to flatter optimization solutions, implying a positive effect on the generalization properties of models optimized under GD.

This work provides a very illuminating perspective on gradient descent through an extremely simple idea. The theoretical grounding for the idea is very sound (although I do have some questions about higher order corrections, see below). Relatedly, I think the "Explicit Regularization" presented (which is stated by the authors not to be the main focus of the paper) could be an influential idea and technique, but more experimentation would probably be necessary before that is likely to happen.

There's not too many weak points of the paper, but the following aspects did come to mind while reviewing. Focusing first on the theoretical parts of the paper, I wonder about the higher order corrections to the error analysis. It seems important to study them and e.g. see whether they have the signs and scaling expected for first correction to be valid widely. The authors do acknowledge that their regularization only holds for "moderate'' learning rates, but they don't attempt to determine what "moderate'' means. What scales do we compare the learning rate to in order to determine whether we're in this regime or not? The coefficient of the first term in the analysis scales with the learning rate and the number of parameters. How do higher order terms scale with the number of parameters? (This matters for understanding how implicit regularization works for larger models.) This sort of analysis seems rather essential and is unfortunately missing. Also, since the toy model is deterministic, there's no way to evaluate generalization error directly -- we only see that the regularization leads to a point on the minima curve with smaller gradient norm. It would be nice if there was a toy that also showed that this improves generalization explicitly, though this is obviously not essential for publishing the paper.

Synthesizing all the information above, I think this paper should be accepted. It is a really nice paper. I think the theoretical claims are broadly correct. I think the empirical methodology is sufficient, e.g. the concept is simple enough and the 2d linear model illustrative of the phenomenon. I think the paper is rather well written. The problem is clearly motivated, the authors explain the phenomenon they intend to study, the analysis of the phenomenon is clear, the toy model is nice and clear, and then confirming experiments follow.

One small criticism relates to the proof of Theorem 3.1, which is the central equation of the paper. The proof starts with a standard result in error analysis, the correction f_1, the derivation of which is in the supplementary material. Since this is so central to the paper, I think it would be nicer if the derivation was at least sketched in the main body of the paper. Perhaps the discussion could be trimmed to make room for such a derivation?


Finally, I have some additional questions and comments:

How does the implicit regularization phenomenon interplay with learning rate schedulers whose goal usually is to decrease the learning rate? Again, this makes me wonder what the intrinsic scale is that we should compare the learning rate to (probably related to the Hessian?) which dictates what the "moderate'' vs. "too small'' vs. "diverging'' regimes really are.

Relatedly, I think it's a well-known fact that the large Hessian eigenvalues end up scaling like 1/(eta*t) over the course of training, with eta the learning rate and t the number of steps. Any comments on how this interplays with the implicit regularization as training progresses?

To my knowledge, usually "implicit regularization'' focuses on _stochastic_ gradient descent as having nice regularization properties. I understand that this paper focuses on only one aspect of the implicit regularization, but comparing to the continuous trajectory seems perhaps to be somewhat of a straw-man. It would be nice to understand how large an effect the regularization is due to discreteness is vs. the regularization due to stochasticity.

With that in mind, how far can the continuous trajectory and the regularized trajectory diverge after t steps? For the toy model, the paths really aren't all that different. If this is just an O(learning-rate) effect and the learning rate has to be small enough to be in the appropriate regime for the Taylor expansion to hold, then how significantly can this really matter in the training of real models? (This certainly motivates the introduction of explicit regularization, but as the authors point out, that's not the main focus of the paper.)

(As a side comment, it's not surprising that for the linear model in section 5 that the implicit gradient regularization is proportional to the L2 regularization of parameters given the relationship between the gradient and the parameter vector for linear models.)

Disclosure: I reviewed this paper for NeurIPS. I read the ICLR submission anew and updated my report accordingly.

---

> ### Author Response · Authors · 2020-11-13
> **Response to review - part 2**
>
> 4. *“how large an effect the regularization is due to discreteness is vs. the regularization due to stochasticity”*
>
> **Response:** Indeed, IGR is not the only source of implicit regularisation, and we believe that stochasticity along with other sources of implicit regularisation are likely to contribute alongside IGR. In the supplementary information we measure the impact of IGR in full batch and stochastic gradient descent, using a variety of different batch sizes and learning rates (Fig. A5, top right). For each batch size, we find that our main result relating the size of IGR and test accuracy holds. This suggests that the impact of IGR is not masked by additional regularisation due to stochasticity. The full backward analysis of the stochastic setting is quite complicated so it falls outside the scope of our paper here, but we will include some discussion on this in our updated paper.
>
> 5. *“Relatedly, I think it's a well-known fact that the large Hessian eigenvalues end up scaling like 1/(eta x t) over the course of training, with eta the learning rate and t the number of steps. Any comments on how this interplays with the implicit regularization as training progresses?"*
>
> **Response:** This seems intuitively consistent with IGR, since IGR puts a pressure on the loss surface slope to be small, preferring minima that can be reached through paths with smaller slopes. Now at a minimum the eigenvalues of the Hessian coincide with the principal curvatures of the loss surface at this point. Our prediction is that higher learning rates encourage (through IGR) the discovery of minima with smaller surrounding slopes, which can be also expressed by saying that the largest eigenvalue of the Hessian should be small at the minima. Furthermore, our backward analysis tells us that the timescale of learning is eta*t, which we refer to as ‘physical time’ (Fig. A4). So yes, it seems natural to expect that the largest eigenvalue of the Hessian to be inversely proportional to the physical time from the point of view of IGR.
>
> 6. *“how far can the continuous trajectory and the regularized trajectory diverge after t steps?”*
>
> **Response:** The error between these is O(t^2), and it can become quite large, especially for large learning rates. The error also depends on the curvature of the loss surface and the initialisation. To see this, in Figure 1, compare the regularized trajectory (red dashed line) and the continuous trajectory (green line).

---

> ### Author Response · Authors · 2020-11-13
> **Response to review - part 1**
>
> Thanks for your review. We are glad that you find our work to be a ‘very illuminating perspective on gradient descent’. In the reply below, we will address your comments in detail, and we will follow up later with corresponding improvements to our submission. In particular, we include a full derivation of the f1 term in the main paper, we will extend the 2d example to demonstrate the generalization benefit of IGR on a simple example as you suggest, and add a discussion on the learning rage range scale.
>
> 1. *“I wonder about the higher order corrections to the error analysis. It seems important to study them and e.g. see whether they have the signs and scaling expected for first correction to be valid widely.”*
>
> **Response:** We provide a recursive formula to define all the higher corrections for gradient descent in proposition A.4 of the appendix. The analysis of these terms is much more complicated than the first order term, and although we have begun to investigate these terms, it is beyond the scope of this paper. In general, the exact size of the higher order terms will depend on the details of a particular loss function. In our experiments, we have calculated the error incurred by not including these higher order terms (in Figure 1; this error is the difference between the red line (exact gradient descent) and the dashed red line (the exact modified flow with the first order correction alone). As you can see, this discrepancy is small. This is more difficult to visualise for higher dimensional models. However, the observation that all our predictions in Section 2 are consistent with experiment strongly suggests that the first order term has a strong impact on deep learning for typical learning rate ranges.
>
> 2. *“The authors do acknowledge that their regularization only holds for "moderate'' learning rates, but they don't attempt to determine what "moderate'' means. What scales do we compare the learning rate to in order to determine whether we're in this regime or not?”*
>
> **Response:** This is a very good question. Some standard results in backward error analysis (see Hairer & Lubish (1997) for instance) indicate that the moderate range lies below $h_0=CR/M$ where $\nabla E$ is analytic and bounded by M in a complex ball of radius R around the initialization point and where C depends on the Runge-Kutta method only, which can be estimated for gradient descent. For each learning rate below $h_0$, we can find an optimal truncation of the modified equation whose flow follows gradient descent very closely, so the higher term corrections are likely to contribute to the dynamics. Given this, we see that the exact value of the upper bound for the moderate regime will correspond to a setting where the optimal truncation is the first order correction only. Calculating this in general is difficult and beyond the scope of this paper. Nonetheless, our experiments strongly suggest that the moderate regime overlaps substantially with the learning rate regimes typically used in deep learning, which are neither very large or infinitesimally small. For our updated submission, we will include this additional discussion.
>
> 3. *“How does the implicit regularization phenomenon interplay with learning rate schedulers whose goal usually is to decrease the learning rate?”*
>
> **Response:** Implicit gradient regularisation can be used to understand the role of learning schedules, since learning rate controls the relative strength of implicit regularisation and loss optimization. For example, for a cyclical learning rate schedule (Smith, 2015), cyclically varying learning rates between large and small learning rates can be interpreted as a cyclical variation between alternate phases of loss optimization and regularization. The most useful amount of regularisation might depend on whether the loss surface is highly curved or plateau-like along some portion of a gradient descent trajectory. For learning rate schedulers with fast changing learning rates, the dynamics of the learning rate scheduler cannot be analysed separately from the dynamics of gradient descent. In such cases, we can again use backward analysis to reveal additional implicit regularisation terms that arise from the learning rate scheduler dynamics. We will add this point to our paper.

---

> ### Author Response · Authors · 2020-11-20
> **Submission updated with new experiments, proof details and clarifications**
>
> Following your suggestions, we have made the following improvements to our paper:
> 1. A new experiment (Fig. A2) to show that gradient regularisation improves generalization in an extension of our two parameter model.
> 2. For Theorem 3.1 we have moved the derivation of the f_1 term to the main text.
> 3. Added Remark 3.2  about results from backward analysis that address your question about learning rate ranges for our backward analysis.
> 4. A new related work section on ‘Learning rate schedules and regimes’ to address your question on this.
>
> If you agree that these changes have improved our submission and addressed your comments we would be grateful if you would consider raising your score.

---

### Official Review · AnonReviewer4 · 2020-10-28
**Recommendation to weak accept on "Implicit Gradient Regularization"**

**Rating:** 6
**Confidence:** 4

**Review:**

The authors show the discrete steps of gradient descent implicitly regularize models by penalizing trajectories that have large loss-gradients, which is called Implicit Gradient Regularization in the paper. The authors adopt a standard argument from the backward error analysis of Runge-Kutta methods to show this phenomenon. In the paper, the authors also provide some empirical results which indicate gradient descent leads to flat minima where test errors are small and solutions are robust to noisy parameter perturbations.

Overall, the paper is marginally above the acceptance threshold from my side. The paper proposes a natural form of implicit regularization when gradient descent is performed to train neural networks. I favor the explanation of the predictions made in the paper and the geometric interpretation of IGR. The experiment results seem to confirm the predictions made in the paper. The idea of backward error analysis is clear and comprehensible.

However, my major concerns are as follows and hopefully the authors can address my concerns in the rebuttal.

First, the paper is lack of existing literature reviews on related work on gradient norm regularization. This may impair the novelty of the paper. In particular, the paper considers explicit gradient regularization which has been discussed and studied for a long time. Moreover, the gradient regularization seems to be very natural in the gradient descent. Since the gradient descent converges by reducing the loss gradient, it seems not clear to me how much effect the additional gradient norm regularization R_{IG} plays in the process. As the authors stated in Prediction 2.2, it is possible that the loss surfaces have nearly equally flat minima. The impact of IGR seems to be unclear in reality. For now, the paper cannot fully convince me of the importance of such implicit regularization.

Second, the main proof techniques are not novel and have been widely used in the area of optimization and machine learning. The paper is sort of lack of technical novelty.

Third, given the sophisticated interactions between different sources of implicit regularization, the paper is lack of careful discussion on other related implicit regularization. The paper only lists some of the discovered implicit regularization in Section 7.

---

> ### Author Response · Authors · 2020-11-13
> **Response to review - part 2**
>
>
> 4. *“Second, the main proof techniques are not novel and have been widely used in the area of optimization and machine learning.”*
>
> **Response:** To the best of our knowledge - and please let us know if you know otherwise - our main theorem (which states that the modified equation at order 1 of a ‘gradient system’ is again a ‘gradient system’) is novel in itself, independent of its application to machine learning. Also, to the best of our knowledge, the use of backward error analysis to quantify inductive biases or implicit regularization effects in the training of overparameterized deep learning models is also novel. It’s a case of extending a well-known method from one field and then applying it to a different problem in another field - a contribution at the interface of two fields. The most closely related work that we are aware of is the work of Qianxiao Li, Cheng Tai, and Weinan E on modified equations for stochastic differential equations, whose mathematical techniques and goals are different, and the very recent work of Guilherme França, Michael I. Jordan, René Vidal - which appeared while we were submitting the first draft of our paper. They use backward analysis ideas to design new optimizers and their focus is on convergence - not on inductive biases. Please do let us know - are we missing any key work using the same approach to quantify implicit regularization or inductive biases as in our paper?
>
> 5. *“Third, given the sophisticated interactions between different sources of implicit regularization, the paper is lack of careful discussion on other related implicit regularization. The paper only lists some of the discovered implicit regularization in Section 7.”*
>
> **Response:** Indeed, IGR is only one of the many possible forms of implicit regularizations that may contribute to training over-parameterized models with gradient descent and we took great care not to overstate the role of IGR and instead, we argue that it is likely to be “just one component of a more complex recipe of implicit and explicit regularization”. Unfortunately, space constraints forced us to drop much of our related work discussion and move some of it to the appendix (Section A.3), but we’d be glad to put this discussion back in using the extra available page. In particular, we see potentially strong relationships with 1) the NTK approach, 2) implicit L2 norm regularization, 3) biases toward large margin, and 4) implicit low-rank biases in deep matrix factorization. For 1), we showed in appendix A.3 that RIG can actually be expressed as the NTK for the least-square loss (Equation A.23). For 2) and 3), we believe some of the previous results obtained in the case of special loss functions such as the least-squares loss or the cross-entropy loss may have an interpretation in terms of RIG, related to the location of the flatter regions for these types of losses (as in our 2d example where the flattest region is located around the origin, making IGR coincides in this case with implicit L2 norm regularization). Is there any other form of implicit regularization or work that you would like us to include in discussion?

---

> ### Author Response · Authors · 2020-11-13
> **Response to review - part 1**
>
> Thanks for your review and for your recommendation that our work should be accepted. We are glad that you find that our ”idea of backward error analysis is clear and comprehensible”. In the reply below, we will address your comments in detail, and we will follow up with corresponding improvements to our submission. If you agree that our responses address your concerns, we would be grateful if you would consider raising your score.  We also have a few questions that will help us to update our paper.
>
> 1. *"First, the paper is lack of existing literature reviews on related work on gradient norm regularization. This may impair the novelty of the paper. In particular, the paper considers explicit gradient regularization which has been discussed and studied for a long time."*
>
> **Response:** We will definitely add a review of literature on explicit gradient regularization (EGR) in our updated submission. We know this literature well, starting with a paper in 1992 by Yan Le Cun on “Double Backpropagation”, up to recent methods for stabilizing GAN training. However, we should make it very clear that EGR is not the focus of our paper. Instead, the novelty of our paper resides in the characterization of a hidden form of implicit regularization in gradient descent (IGR) that was not previously known. Our EGR experiments simply act as control study in our work, to demonstrate that the R_IG term arising implicitly in gradient descent can indeed improve test accuracy independent of any possible confounding effects that may arise when we control IGR implicitly through the learning rate. Namely, if we couldn’t observe a significant boost in model test accuracy by adding the RIG term explicitly, our prediction that implicit regularisation helps to boost test accuracy would have been in doubt. Also, the fact that explicit gradient regularisation has been used successfully in past experimental work is a further argument for the utility of the implicit regularization that we have found hidden in gradient descent flows, and which our paper uncovers for the first time, to the best of our knowledge.
>
> 2. *“Since the gradient descent converges by reducing the loss gradient, it seems not clear to me how much effect the additional gradient norm regularization R_{IG} plays in the process.”*
>
> **Response:**  This is an excellent question, and is something that we did a lot of work to address, since the size of R_IG clearly depends on the size of gradients which do become small during gradient descent.  Indeed, for infinitesimally small learning rates, IGR has no impact. However, for larger learning rates such as those typically used in deep learning, Theorem 3.1 proves that IGR has an impact. IGR does not change the optima of our loss (since all optima have a zero loss gradient). Rather, IGR changes the trajectory of gradient descent. Instead of following the steepest gradient exactly, gradient descent steps off the path of the exact gradient flow onto a shallower path. This is most clear in Figure 1, where we see that the modified flow with IGR follows a shallower path. The cumulative impact of this gradient regularisation is that gradient descent typically moves toward flatter regions of the loss surface. For higher dimensional models, such as neural networks, we cannot plot the full parameter space. Instead, we make several falsifiable predictions (in Section 2) and we find that all of these predictions hold in our deep learning experiments. In particular, from Theorem 3.1 we can predict that IGR encourages smaller values of R_IG relative to the loss E (Prediction 2.1). We confirm this prediction in Figure 2 and Figure A2c.  We control for the possibility that larger learning rates are simply progressing more quickly, by comparing trajectories at the time of maximum test accuracy (Fig 2), and as a control, we compare trajectories throughout training using rescaled physical time (Fig A4) using large compute budgets. We also predict and confirm that higher learning rates increase test accuracy (Fig 2), and have flatter solutions at the time of maximum test accuracy (Fig 3a). We will work to improve our explanation of this in our submission. If you have any further questions related to this central point, please let us know and we would be glad to clarify.
>
> 3. *“As the authors stated in Prediction 2.2, it is possible that the loss surfaces have nearly equally flat minima. The impact of IGR seems to be unclear in reality”*
>
> **Response:** Although it is possible to deliberately construct a loss surface in such a way that IGR will not have any impact, this is not the case for widely used loss surfaces, such as overparameterized deep neural networks (The Mexican-hat loss example we mentioned is somewhat contrived). In particular, in our deep learning experiments we show that larger learning rates do indeed have flatter minima (Fig 3a, and Fig A.6c) and in our 2d model we clearly see that IGR leads to flatter minima (Fig 1).

---

> ### Author Response · Authors · 2020-11-20
> **Submission updated with new experiments, proof details and clarifications**
>
> Following your suggestions, we have made the following improvements to our paper:
> 1. Added Remark 4.1 describing related work on EGR.
> 2. Extra related work sections on the relationship between IGR and other forms of implicit regularisation including implicit weight norm regularisation and low rank regularisation.
> 3. New related work section - ‘Neural Tangent Kernel’ - on the exact mathematical comparison between IGR and NTK methods.
> 4. New related work section on ‘Learning rate schedules and regimes’
> 5. New related work discussion of Runge-Kutte methods including backward analysis.
>
> If you agree that these changes have improved our submission and addressed your comments, we would be grateful if you would consider raising your score.

---

> > ### Comment · AnonReviewer4 · 2020-11-24
> > **Appreciate the effort in addressing my concerns**
> >
> > I really appreciate the effort put by the authors in addressing my concerns and updated version looks good to me. Since I hesitated over the score between 5 and 6 before, based on the authors' responses and the updated manuscript I decided to keep my evaluation at 6 i.e. "marginally above acceptance threshold".

---

> > > ### Author Response · Authors · 2020-11-24
> > > **Thanks for your review**
> > >
> > > We are glad that you believe our "updated version looks good".
> > >
> > > In your review, you comment on our literature review. Is there some additional literature that you would like us to discuss, or some additional experiments or analysis that might help to further strengthen our paper?
> > >
> > > Thanks again.

---

### Official Review · AnonReviewer3 · 2020-10-28
**Nice mathematical analysis of gradient descent trajectory, but limited impact and improvable readability**

**Rating:** 6
**Confidence:** 4

**Review:**

## Edit after rebuttal

I have updated my evaluation (from 4 to 6) based on the changes made in the manuscript and the responses by the authors. See details in my response to the authors: https://openreview.net/forum?id=3q5IqUrkcF&noteId=qZzolK7E-wF

## Summary of the paper

This article analyses the discrepancy between the trajectory of the discrete numerical implementation of (full) batch gradient descent and the trajectory of the continuous gradient flow, for optimising over-parameterised models. The authors use backward error analysis to derive a modified loss whose gradient flow that better approximates the actual trajectory of gradient descent. The modified loss is the original loss plus a term that is proportional to the second moment of the loss gradients, the learning rate and the model size. The authors call this term "implicit gradient regulariser" and claim that it encourages flatter minima, higher test accuracy and more robust optima.

## Summary of merits and concerns

### Merits

+ The paper presents a compelling mathematical analysis of the discrepancy between the actual trajectory of gradient descent and the gradient flow that determines it. The discrepancy is due to the discretisation of the gradient flow by a step size determined by the learning rate. This analysis sheds some light on why larger learning rate provides better results.
+ The mathematical derivations provided by the authors may be used by other researchers to better understand the dynamics of gradient descent.
+ The sections of the supplementary material concerned with mathematical proofs and derivations (I have more carefully read A.1 and A.2) are very clearly presented
+ The authors identify a good number of related articles.

### Concerns

- In my opinion, the authors overstate the role of the discrepancy between the gradient descent trajectory and the gradient flow in explaining the generalisation properties of over-parameterised models.
- The language used is confusing and contributes to the overstatement. For example, the authors state that "implicit gradient regularisation" "encourages", "guides", "has an impact on", etc., while it would probably be more accurate to say that it may help "explain" (some properties of gradient descent dynamics).
- Despite the rigour of the mathematical derivations, to the best of my understanding, the results do not fully support the claims that higher accuracy, flatter minima and better robustness are encouraged by implicit gradient regularisation, since other relevant aspects are not taken into account, such as parameter initialisation, model parameterisation, etc. In the same line, the experimental setup leaves important questions open and lacks transparency.
- There is substantial room for improvement regarding the readability of the paper. Examples are an overwhelming amount of cross-references back and forth between equations, figures, predictions, sections of the appendices, etc. that make it hard to navigate the paper; many of these cross-references are made before being introduced; repeated use of chains of multiple citations that certainly hinder the readability of some paragraphs.

## Evaluation and justification

While I acknowledge the merits and contributions of the paper---in particular the potential of the mathematical results for helping us understand the dynamics of gradient descent and, in a broader sense, the usefulness of the methods used here for studying machine learning problems---I contend that the theoretical and empirical results fall short at supporting the rather ambitious claims and that, at its current state, the impact of the paper is limited. These reasons, as well as my concerns about the clarity of the presentation and readability, led me to lean towards the recommendation of rejection. In the remaining part of my review I will discuss in more depth these concerns in order to better justify my recommendation and with the intention to provide constructive feedback for potential subsequent work on the paper.

### Overstated role of "implicit gradient regularisation"

The discrepancy in the trajectory of gradient descent with respect to the gradient flow is an intrinsic property of the method and a direct consequence of using finite step sizes, that is a positive learning rate. With an infinitesimal learning rate, the trajectory would perfectly approximate the gradient flow and, as a consequence, the optimisation could get stuck at the first critical point (for instance a suboptimal local minimum) after initialisation, in the case of non-convex loss landscapes. In other words, as it is well-known, a sufficiently large learning rate is required to escape local minima---the authors are of course aware of this simple and well-known notion and do mention it in the paper. Stating that implicit gradient regularisation "encourages" higher test accuracy and flatter optima, and "guides" gradient descent along shallower paths is an unnecessarily complicated way of putting the simple fact that gradient descent would not work if it followed the continuous gradient flow in infinitesimal steps. This said, I do acknowledge the rigour of the mathematical derivations and the fact that the results might be used to partly explain the dynamics of gradient descent.

In my opinion, one potential application of the backward error analysis of gradient descent presented in this paper is understanding the optimisation dynamics when the learning rate is larger. However, the analysis should be very careful and perhaps limited to the earlier stages of training, rather than focusing on the characteristics of the landscape at the end of the optimisation process, as it is analysed in the paper. A large learning rate actually prevents the algorithm from reaching better optima and in fact a common practice in neural network training is to decay the learning rate when the loss plateaus. Decaying the learning rate in turn anneals the "implicit gradient regulariser" $R_{IG}$ analysed in this paper, and brings the trajectory of gradient descent closer to the gradient flow. Implicit gradient regularisation would therefore fail to explain the characteristics of the loss landscape at the end of training if the learning rate is decayed, as is done in many successful models.

The optimisation trajectory that gradient descent follows with larger learning rate in over-parameterised models is an interesting subject of study and it may be analysed through backward error analysis as this paper proposes. However, in my opinion, this paper overstates the role of implicit gradient regularisation. I argue that the language used to state the claims exagerates the findings, since many other factors that play a role are not considered. Some examples are the following:

- "We find that IGR can account for the observation that learning rate size is correlated with test accuracy and model robustness" (Section 1)
- "EGR produces a boost of more than 12% in test accuracy" (Section 6)
- "networks with small learning rates or fewer parameters or both will have less IGR and worse test error" (Section 8)
- "solutions with high IGR are more robust to parameter perturbations" (Section 8)
- "IGR produces minima with low test error" (Section 8)

In sum, the main contribution of the paper seems to be Section 3, which shows "that gradient descent follows the gradient flow of the modified loss $\tilde{E}$ more closely than that of the original loss $E$". While this is a correct conclusion and the mathematical derivations are rigorous (though not very clearly presented), the relevant aspect is the implications of this on the dynamics and the relationship with the learning rate, which should be very carefully analysed as it is the cause of the discrepancy in the trajectories but it can lead to deficient optimisation. I believe that an interesting avenue is to analyse the role of larger learning rates on gradient descent dynamics, potentially through backward error analysis as in this paper, at early phases of training, following, for instance, on the work by Li, Wei and Ma (2019). This paper briefly outlines some ideas in this regard at the end of Section 3, but the focus diverges in the rest of the paper and the experimental results do not shed much light either (as discussed below). In my opinion, putting more focus on this aspect would make the paper stronger.

### Concerns about the experimental setup

In the experiments presented in the paper, the authors analyse $R_{IG}$ and the test accuracy "at the time of maximum test accuracy for each network that fits the training data exactly". However, we should note the following: First, the models trained on MNIST may achieve 100 % train accuracy, but as we can see in Figure A.2 of the appendix, the models do not achieve zero loss and there are differences depending on the learning rate. The authors also perform experiments on CIFAR-10, a more challenging task, but neither the training loss nor the training accuracy are reported in this case. It is claimed, in the case of CIFAR-10 in the appendix in Figure A.6, that "the test accuracy increases and the regularization value $R_{IG}$ decreases as the learning rate increases", but this could be simply explained because the models trained with smaller learning rate are simply not sufficiently optimised, with even lower training accuracy and loss. I would like to note that this denotes a relevant lack of transparency in the report of the results.

Similarly, from Figure 2a, the authors claim that $R_{IG}$ is smaller for larger learning rate x network size. While this generally holds across variation within each model, that is for increases in learning rate but not network size (the comparison highlighted in the figure), the conclusion is less clear for variation within each learning rate, that is for increases in network size but not learning rate.

Another claim is that models trained with higher learning rate are more robust to parameter perturbation. This is an interesting observation, but the paper does not demonstrate that this is due to implicit gradient regularisation. The claim is derived in part from the claim that the end region in the loss landscape is flatter, whose connection with implicit gradient regularisation is problematic for the reasons stated above. Otherwise, the claim is derived from the experimental setup. However, only a few experiments are performed in this regard. For example, why is this only analysed on MNIST, but not on CIFAR-10 models? Furthermore, from the experiments on MNIST with various learning rates (Figures 3a and 3b), the authors conclude that "neural networks with larger learning rates, and hence, with stronger [implicit gradient regularisation] have smaller slopes at the time of maximum test accuracy" and "are more robust to parameter perturbations after training". However, how do the authors control that this is not due to the smaller training loss (Figure A.2) achieved with larger learning rate, as well as the longer effective (physical) training time (Figure A.4)?

### Clarity

Although there some parts of the paper that clearly presented, there are a few aspects that make some sections hard to read. I humbly believe that there is significant room for improving the clarity of the paper. I will use some specific examples to illustrate my concerns regarding clarity and readability:

- There are too many cross-references throughout the whole paper. The authors very often refer to equations, figures, predictions, theorems, etc. to illustrate their points. While this can be useful at times, if abused it runs the risk of making the paragraphs hard to understand and the paper difficult to navigate, at least in my case. This especially true if the references have not been introduced yet (often the case in sections 1 and 2), are several pages below (Figures 1 and 2 in pages 5 and 6 and referenced in page 2) and there are multiple pointers to the appendix. Take as an example the last paragraph in page 6: in one single paragraph there are 11 references to different parts of the paper. The paragraph, judging for the first sentence, should introduce the experiments on CIFAR-10. However, if one aims to understand and interpret the results on CIFAR-10, they would need a considerable effort and time to scrutinise the paper, especially several sections in the appendices, to finally conclude that the results are not reported with enough transparency. This lack of clarity not only makes the paper harder to read, but the confusion also hinders its transparency.
- In some paragraphs, there are so many citations that it is hard to read the sentences. Take, for instance, the first paragraph in Section 7. Probably more than half of the characters correspond to citations, with chains of up to 11 citations in one go and words ("stochasticity") hidden in between chains of citations. Is it really necessary to cite all these papers? If so, is this the best possible way in terms of clarity? Is the reader suppose to read all these papers? Some have referred to this as ["shotgun citations"](http://robertposhea.blogspot.com/2013/05/shotgun-citation.html).
- Long mathematical derivations in-line within a paragraph are also hard to follow. The "clearest" example of this is the first paragraph in page 4. I would like to note that, in contrast, appendices A.1 and A.2 are very clear and well written, perhaps because the mathematical expressions are developed vertically with pertinent explanations.
- The legend of some figures is very small and therefore hard to read. See, for instance, Figure 1a.

## Questions and suggestions

Below I will list some questions or suggestions that occurred to me while reviewing the article. These have had a small impact in my assessment of the paper but the authors may find them useful for future work on this or subsequent papers.

- As the authors acknowledge, one limitation of the present work is that it studies full batch gradient descent, while the optimisation workhorse in deep learning is stochastic gradient descent (SGD). I understand this limitation, but I think the paper would be stronger if it outlined some ideas for future work, perhaps related to previous work that has addressed SGD [1, 2].
- Another common practice in training deep neural networks is the decay of the learning rate. As discussed above, it is not obvious how implicit gradient regularisation would explain the benefits of learning rate decay and how these would be connected with the claims in the paper. DO the authors have any thoughts in these regards?
- "EGR produces a boost of more than 12% in test accuracy" (Section 6): While this seems impressive, a closer look reveals that the increase is from 62 to 74 % accuracy on CIFAR-10, that is well below the performance of typical architectures. By way of illustration, All-CNN (2014) achieves more than 90 % accuracy. This observation made me pay less attention to EGR and not really take it into account for my assessment, since the authors state that this is not the focus of the paper. However, since it is indeed included and discussed in the paper, I contend that the experiments should be more rigorous and the claims more careful.
- Inspired by EGR though, would it be an idea to consider an alternative loss of $E - R_{IG}$ (or along this line) so as to analyse a model that explains away the contribution of $R_{IG}$, for comparison theoretically or experimentally? I would need to think about this more carefully and see if it would make sense at all, since the discretisation would also affect this new loss.

## Minor comments and potential typos identified

The following list collects some minor comments or typos that I have identified. These are intended to improve the quality of the paper and did not have a substantial impact on my evaluation of the paper. The list is not organised into any particular order.

- The title may be too broad and slightly ambiguous.
- It seems that the authors used [hyphens](https://en.wikipedia.org/wiki/Hyphen) for parenthetical phrases, where an [em dash](https://en.wikipedia.org/wiki/Dash#Em_dash) would be probably more correct. Note that this occurs several times throughout the paper, but one example is in the first sentence of the introduction (Section 1).
- The meaning of variable $m$ is not explained in Equations 3 and 4.
- In Section 3, for better clarity, the authors may like to develop the acronym ODE (ordinary differential equation)
- In page 4, consider using "cancel out" instead of "kill"
- Typo: "we will _demonstration_ the effectiveness of EGR"
- Consider spelling "CIFAR-10" with upper case in page 4, as is commonly written, instead of "Cifar-10". Also for consistency throughout the paper.
- "are more robust to parameter perturbations after training (_Figure 3c_)" (page 6): I believe it should read "Figure 3b".
- In Section 7, considering mentioning data augmentation as an implicit source of implicit regularisation which is very commonly used in practice.
- Some relevant papers that the authors may consider discussing are [3, 4]
- It is not clear if the first paragraph in page 3 corresponds to Prediction 2.4, or it is a new paragraph.
- Theorem 3.1 refers back to Equation 2, while it would be clearer to explicitly express it there.
- Typo: "is a global _minima_" should probably read _minimum_
- The first paragraph of page 6 is really long and hard to read.


## References

[1] Alnur Ali, Edgar Dobriban, and Ryan Tibshirani. The implicit regularization of stochastic gradient flow for least squares. arXiv:2003.07802, 2020.

[2] Qianxiao Li, Cheng Tai, and Weinan E. Stochastic modified equations and adaptive stochastic gradient algorithms. In International Conference on Machine Learning, volume 70, pp. 2101–2110, 2017.

[3] Gidel, Gauthier, Francis Bach, and Simon Lacoste-Julien. "Implicit regularization of discrete gradient dynamics in linear neural networks." Advances in Neural Information Processing Systems. 2019.

[4] Saxe, Andrew M., James L. McClelland, and Surya Ganguli. "Exact solutions to the nonlinear dynamics of learning in deep linear neural networks." arXiv preprint arXiv:1312.6120, 2013.

---

> ### Author Response · Authors · 2020-11-13
> **Response to review - part 3**
>
> 11. *“There is substantial room for improvement regarding the readability of the paper.”*
>
> **Response:** Thanks for pointing us to ways to improve clarity as well as typos. We will incorporate them in our updated submission.
>
> 12. *“I think the paper would be stronger if it outlined some ideas for future work [in SGD]”*
>
> **Response:** Although SGD is beyond the scope of this paper, it is possible to extend backward analysis to calculate implicit regularisation in SGD. Although this calculation is quite involved and deserves full treatment as a separate paper, we will include some discussion in this paper on backward analysis of SGD.  Also, in our experiments, we calculated R_IG in SGD using a variety of different batch sizes and learning rates (Fig. A5, top right) and found that for each batch size, our main result relating the size of R_IG and test accuracy holds.
>
> 13. *“it is not obvious how implicit gradient regularisation would explain the benefits of learning rate decay and how these would be connected with the claims in the paper. DO the authors have any thoughts in these regards?”*
>
> **Response:** Implicit gradient regularisation can be used to understand the role of learning schedules, since learning rate controls the relative strength of implicit regularisation and loss optimization. For example, for a cyclical learning rate schedule (Smith, 2015), cyclically varying learning rates between large and small learning rates can be interpreted as a cyclical variation between large and small amounts of IGR (i.e., alternate phases of optimization and regularization). We also believe that IGR is most useful at the beginning of training orienting the search toward flatter regions at this stage, and less important in later stages of the training, when a flatter region has been reached, and where now the convergence to any of the flatter minima is then more important at this latter stage of the training. This is consistent with the work of Li, Wei and Ma (2019) that associates generalization with larger learning rate at the beginning of training. This is also consistent with common learning rate decay schedules reducing the learning rate as training progresses. It’s also possible to analyse the additional effect of a learning rate decay schedule through backward analysis, although it’s beyond the scope of our paper. It’s very possible that different learning rate schedules introduce different inductive biases.
>
> 14. *"EGR produces a boost of more than 12% in test accuracy" (Section 6): While this seems impressive, a closer look reveals that the increase is from 62 to 74 % accuracy on CIFAR-10, that is well below the performance of typical architectures. By way of illustration, All-CNN (2014) achieves more than 90 % accuracy. This observation made me pay less attention to EGR and not really take it into account for my assessment, since the authors state that this is not the focus of the paper. However, since it is indeed included and discussed in the paper, I contend that the experiments should be more rigorous and the claims more careful.”*
>
> **Response:** Our EGR experiments simply act as control study in our work, to demonstrate that the R_IG term arising implicitly in gradient descent can indeed improve test accuracy independent of any possible confounding effects that may arise when we control IGR implicitly through the learning rate (such as implicit regularisation from sources of stochasticity for instance). Namely, if we couldn’t observe a significant boost in model test accuracy by adding the RIG term explicitly, our prediction that implicit regularisation helps to boost test accuracy would have been in doubt. We are not attempting to argue that EGR alone can produce state-of-the-art results on CIFAR. We will make this clearer in our paper update.
>
> 15. *“Inspired by EGR though, would it be an idea to consider an alternative loss of E−RIG (or along this line) so as to analyse a model that explains away the contribution of RIG for comparison theoretically or experimentally? I would need to think about this more carefully and see if it would make sense at all, since the discretisation would also affect this new loss."*
>
> **Response:** Yes, the discretization would also affect the RIG term in E-RIG and the modified loss in this case would not yield back the original loss. So gradient descent on E - RIG would not coincide with the exact gradient flow of E.  It’s easier to do the reverse to show the existence of the effect experimentally as we do in example 2d, where we perform gradient descent with very small learning rate on E + hRIG and observe that this flow follows closely that of  gradient flow performed on E with learning rate h.

---

> ### Author Response · Authors · 2020-11-13
> **Response to review - part 2**
>
> 5. *“The language used is confusing and contributes to the overstatement. For example, the authors state that "implicit gradient regularisation" "encourages", "guides" ...”*
>
> **Response:** The language we have used here is the language of regularisation theory. This was carefully chosen to describe the effect of regularization as a mechanism that "puts a pressure" on the optimization path to "encourage" certain types of solutions or behavior. We don’t believe that this overstates the impact of IGR, but rather, it is an accurate characterisation of the IGR. We do make definitive statements in our theorem, showing that there exists an IGR term. However, during gradient descent, there may be obstacles on the path preventing gradient descent from reaching flatter regions in spite of this IGR term. That’s why we use verbs such as “encourages”, “guides”, or “helps”  instead of “leads to”,  “produces”, or “results in”. We hope this clarifies our intent behind the use of this intuitive language.
>
> 6. *“relevant aspects are not taken into account, such as parameter initialisation, model parameterisation, etc. ”*
>
> **Response:** We do control for parameter initialisation and model parameterisation. Specifically, we use random initialisation for each gradient descent trajectory, and, for our MNIST experiments, we use a range of networks with different sizes and we find that our results are not sensitively dependent on these choices. This is not surprising because Theorem 3.1 holds regardless of initialization and parameterisation.
>
> 7. *“the models trained on MNIST may achieve 100 % train accuracy, but as we can see in Figure A.2 of the appendix, the models do not achieve zero loss and there are differences depending on the learning rate.“*
>
> **Response:** The models will never reach zero loss, since it is a cross entropy loss, but this is not a problem since accuracy is a more important metric of performance. For this reason, we compare models at the time of maximum test accuracy. As a control, we also compare models after an equal (large) number of step sizes (Fig. A2) and after an equal (large) number of physical time steps (Fig A4). In each case, our main prediction 2.1 holds.
>
> 8. *As requested, we will provide training loss and training accuracy for CIFAR-10*
>
> 9. *“[Cifar] models trained with smaller learning rate are simply not sufficiently optimised”*
>
> **Response:** Each model is optimised and successfully achieves maximum test accuracy (Fig A6a). We use large amounts of compute, up to 50000 epochs, to train our models well beyond the time that they reach maximum test accuracy.
>
> 10. *“the authors claim that RIG is smaller for larger learning rate x network size … the conclusion is less clear for variation within each learning rate, that is for increases in network size but not learning rate.”*
>
> **Response:** In Figure A5 (top, middle panel), it is clear that RIG is smaller for increases in network size.

---

> ### Author Response · Authors · 2020-11-13
> **Response to review - part 1**
>
> Thanks for your thorough review. We are glad that you find that our paper “presents a compelling mathematical analysis”. We also appreciate the detailed and constructive feedback that will allow us to improve our paper substantially. We have attempted to address all of your comments, but please do let us know if there are any particular topics you would like to discuss in further detail. If you agree that our responses address your concerns, we would be grateful if you would consider raising your score.
> 1. *“Nice mathematical analysis of gradient descent trajectory, but limited impact”*.
> **Response:**  We respectfully disagree that our work has limited impact. In this paper, we quantify a new form of implicit regularization in gradient descent, which is impactful because gradient descent, and its variants are so widely used in machine learning. Also, our experiments indicate that implicit gradient regularisation plays an important role in deep learning, which is impactful because deep learning is widely used, and not yet fully understood. Indeed this has been a very active area of research in recent years. Furthermore, we have developed the use of backward error analysis as a general method for quantifying inductive biases, which we  believe will become a standard tool for understanding other widely used optimisers, such as Adam for instance, in future.
>
> 2. *“the authors overstate the role of the discrepancy between the gradient descent trajectory and the gradient flow in explaining the generalisation properties of over-parameterised models.”*
>
> **Response:** In our paper, we write that IGR is likely to be “just one component of a more complex recipe of implicit and explicit regularization” and we provide extensive references to alternative sources of implicit regularisation. As such, we do not agree that we have overstated the role of IGR. Furthermore, our generalisation experiments all support our view that IGR does help to improve test errors. In particular, our EGR experiments clearly show that  the R_IG reduces test error (Figure 3c) and our full batch experiments provide definitive examples of cases where stochasticity cannot account for improvements in generalisation (Figure A5).
>
> 3. *“As it is well-known, a sufficiently large learning rate is required to escape local minima---the authors are of course aware of this simple and well-known notion and do mention it in the paper. Stating that implicit gradient regularisation "encourages" higher test accuracy and flatter optima, and "guides" gradient descent along shallower paths is an unnecessarily complicated way of putting the simple fact that gradient descent would not work if it followed the continuous gradient flow in infinitesimal steps.”*
>
> **Response:** We believe there is a misunderstanding here of the impact of IGR. We do not think that IGR alone helps much in escaping from local minima, especially in full batch gradient descent. The learning rate regime allowing to escape from local minima by “jumping” over them is beyond the moderate regime where the first order modified equation approximates the gradient descent dynamics. Namely, for the same reason that gradient descent with very small learning rate may be trapped in local minima of the original loss, gradient descent with moderate learning rate will also remain trapped in local minima of the modified loss, which are not that different. IGR helps select in the submanifold of global minima, those solutions that are located in regions surrounded by flatter slopes. We should make this point very clear in our paper. Thanks for raising it.
>
> 4. *“Decaying the learning rate in turn anneals the "implicit gradient regulariser" RIG  analysed in this paper, and brings the trajectory of gradient descent closer to the gradient flow. Implicit gradient regularisation would therefore fail to explain the characteristics of the loss landscape at the end of training if the learning rate is decayed, as is done in many successful models.”*
>
> **Response:** We believe that IGR is most useful at the beginning of training orienting the search toward flatter regions at this stage, and less important in later stages of the training, when a flatter region has been reached. This is consistent with common learning rate schedules reducing the learning rate as training progresses. This is also consistent with the work of Li, Wei and Ma (2019) that associates generalization with larger learning rate at the beginning of training.

---

> ### Author Response · Authors · 2020-11-20
> **Submission updated with new experiments, proof details and clarifications**
>
> Following your suggestions, we have made the following improvements to our paper:
>
> 1. Added Remark 3.4 explaining how our results demonstrate that IGR does not involve escape from local minima
> 2. Extra experiments (Fig A7), as requested, showing that our models trained on CIFAR are fully optimised.
> 3. Extra detail on EGR in section 4, explaining that we do not propose EGR as a new method, but rather, as a control experiment for IGR.
> 4. Added the additional references you listed.
> 5. Improved the clarity of our writing throughout the text.
>
> If you agree that these changes have improved our submission and addressed your comments we would be grateful if you would consider raising your score.

---

> > ### Comment · AnonReviewer3 · 2020-11-22
> > **Updated score due changes and comments. Some concerns remain.**
> >
> > I truly appreciate the effort put by the authors in addressing my concerns and explaining some points I raised. I also very positively value the review of the manuscript, particularly regarding the clarity of the presentation and the inclusion of some results that were omitted in the submitted version in detriment of the transparency of the experiments. Based on the authors' responses and the updated manuscript I reassessed my evaluation up to "marginally above acceptance threshold".
> >
> > The clarity of the paper has improved and so has the transparency of the results, even though one has to carefully read the appendices to get the full picture of the experiments. Therefore there is still room for improvement in this regard. Some of my concerns have been mitigated thanks to the responses of the authors and the updated manuscript. I now recommend acceptance, though marginally, because I see the value in the mathematical analysis of gradient descent proposed in the paper, and it may inspire future research to use similar tools to further analyse the dynamics of gradient descent in overparameterised models. Still, my concerns about the overstatement of some claims and the language used remain.
> >
> > I believe that the conclusions from the mathematical analysis could have been presented in a less confusing way, making the paper stronger, more accessible and more insightful to our understanding of gradient descent. In my review, I argued that the language used has an impact in this confusion and my opinion has not changed in this regard. The authors responded that the language they have used is that of regularisation theory. However, here we are not discussing a regularisation term that is added to a baseline objective, but rather about the properties of one method, gradient descent, which the authors analyse through the decomposition of the loss into a "modified loss" and a "regularizer" $R_{IG}$. This decomposition is a direct consequence of the discretisation of the optimisation process, which is governed by the step size, that is the learning rate. Hence, I think that the analysis presented in this paper does shed light on how larger learning rates (at the early stages of the optimisation process) may encourage more favourable optimisation trajectories. In contrast, the language used is rather that "implicit gradient regularisation encourages/guides/etc." What can be controlled and has a great impact in deep learning is the learning rate and it is important to understand the causal effect. I do not question the rigour of the language, but it makes the arguments confusing, in my opinion, limiting the impact, together with other concerns argued in my original review. One example is "Prediction 2.1. IGR encourages smaller values of $R_{IG}(θ)$". What is the cause and effect here? Are not IGR and $R_{IG}$ the same? Would it not be less confusing and more precise to say that "higher learning rates (within a moderate range) encourage X/Y/X/..."? This questions arise to me through the paper. By way of illustration of the opposite, a related paper that also aims to better understand the dynamics of gradient descent in the early phase of optimisation and the role of (larger) learning rate (in stochastic gradient descent though) is (Jastrzebski et al., ICLR 2020). In that paper, the contributions and conclusions are more clear to me because the cause and effect are less confusing.
> >
> > On a different note, although the new figures included in the appendices help better understand the results, some of my concerns regarding the experimental setup still remain.
> >
> > Finally, I found another potential typo in "Our core idea, is [...]" (incorrect comma between subject and verb) and the labels in Figure A.7 are cut by the figures.
> >
> > ---
> >
> > Jastrzebski et al. The break-even point on optimization trajectories of deep neural networks. ICLR 2020.

---

> ### Author Response · Authors · 2020-11-24
> **Thanks for increasing your score following our paper updates**
>
> We are glad that our updates and comments have largely addressed your concerns and that you believe our work “may inspire future research”.
>
> For the outstanding issue that you raise, regarding confusion in our use of language, we now understand the concern. There is confusion about our use of the term IGR (e.g “IGR encourages smaller values of R”). You point out that smaller values of R are caused by increased learning rates, whereas we write that it is caused by IGR. When we write that ‘IGR causes X’, indeed, we do mean that learning rate increases cause X, but we are also referring to other mechanisms related to the discretisation of gradient descent. In particular, we identify network size along with learning rate as an important factor (e.g. in Figure 2 we see that learning rate x network size encourages smaller values of R and higher test accuracy). This confusion has probably arisen because we did not clearly define “IGR” in the text.
>
> To address this, we have now included a new Definition 2.1.
>
> We have also included the additional reference that you suggest along with additional discussion in the related work section on the complementary relationship between our backward analysis and the break-even point.
>
> Thanks again for your careful review, which we believe has led to worthwhile improvements in the clarity of our text. If you agree that this important clarification has improved our submission and would like to see our paper accepted, we would be grateful if you would consider raising your score.

---

> > ### Comment · AnonReviewer3 · 2020-11-24
> > **Definition 2.1 is a good idea**
> >
> > I think that the inclusion of an explicit definition of implicit gradient regularization - the title of the paper - will certainly help future readers. I had missed this in the original submission. I thank the authors for addressing this point.
> >
> > I am pleased to read that the authors considered my comments useful for their paper.

---

### Author Response · Authors · 2020-11-13
**Thank you for the positive reviews and constructive feedback.**

Thanks to our reviewers for their thorough review of our work. We are grateful for the kind comments -  ‘very illuminating perspective on gradient descent’, ”clear and comprehensible, “ a compelling mathematical analysis”.  We also appreciate the detailed and constructive feedback that will allow us to make improvements to our paper, and hopefully help to resolve points of disagreement between the reviewers. In particular, we note that all the concerns raised can be resolved with improvements to the clarity of our writing, along with additional experiments. We will provide these in an updated paper submission.

---

### Decision · Program_Chairs · 2021-01-07
**Final Decision**

**Decision:**

Accept (Poster)

**Comment:**

The paper presents a mathematical analysis of the discrepancy between GD and GF trajectories. Following the discussion period, a knowledgeable R3 updated his/her initial rating from 4 to 6,  a knowledgeable R4 raised his/her score from 5 to 6. Finally, a very confident R1 considers this a good paper that should be accepted. He/she indicates that this paper provides a unique and very illuminating perspective on gradient descent through an extremely simple idea. The topic is very timely. I agree with R1 that the paper contributes a refreshing perspective with important elements which should be of interest to a good number of researchers. Taking into account the discussion, confidence levels and ratings of the reviewers, I am recommending the paper to be accepted. I would like to encourage the authors to take the reviewers' comments carefully into consideration when preparing the final version of the article.